# Dietary Supplementation of *Auricularia auricula-judae* Polysaccharides Alleviate Nutritional Obesity in Mice via Regulating Inflammatory Response and Lipid Metabolism

**DOI:** 10.3390/foods11070942

**Published:** 2022-03-24

**Authors:** Qian Liu, Ruisen Ma, Si Li, Yujie Fei, Jing Lei, Ruoyu Li, Yu Pan, Sining Liu, Langhong Wang

**Affiliations:** College of Food Science and Technology, Northwest University, Xi’an 710069, China; 2019120033@stumail.nwu.edu.cn (R.M.); 2019120011@stumail.nwu.edu.cn (S.L.); 2020120008@stumail.nwu.edu.cn (Y.F.); 2019120004@stumail.nwu.edu.cn (J.L.); 2019120009@stumail.nwu.edu.cn (R.L.); panyui@stumail.nwu.edu.cn (Y.P.); liusining@stumail.nwu.edu.cn (S.L.); wlhong@nwu.edu.cn (L.W.)

**Keywords:** *Auricularia auricula-judae* polysaccharide, nutritional obesity, high-calorie diet, lipid metabolism, inflammatory response

## Abstract

The incidence of lipid metabolism disorder and obesity that is caused by high-calorie diets is increasing year by year, which has become an urgent global health problem. This study was performed to explore the intervention effects of polysaccharides that were extracted from *Auricularia auricula-judae* resources in the Qinba Mountain area on nutritional obesity in C57BL/6J mice that was induced by high fat and high fructose diets (HFFD) and to investigate their underlying molecular mechanisms. The results showed that dietary supplementation of *Auricularia auricula-judae* polysaccharides (AAP) significantly improved mice’s insulin resistance state, altered serum lipid metabolites, and slowed down body weight gain that was induced by HFFD. In addition, AAP supplementation decreased inflammatory factor levels and alleviated liver histomorphology changes. Furthermore, AAP down-regulated liver adipogenic-related gene expressions, suppressed cholesterol synthesis-related gene levels, up-regulated fatty acid β-oxidation-related gene expressions, and promoted cholesterol efflux-related gene expressions, thus improving mice hepatic lipid metabolism homeostasis. Moreover, the intervention effects were closely related to mitochondrial function. These results provide a scientific basis for the further development and utilization of *Auricularia auricula-judae* resources in the Qinba Mountain area.

## 1. Introduction

With the continuous improvement of living standards, the dietary structure of residents has undergone tremendous changes. Obesity, insulin resistance, type II diabetes, and other metabolic syndromes that are caused by glucose and lipid metabolism disorders have become global health problems [1]. According to the World Health Organization, the number of obese people in the world surpassed 650 million in 2016, with a projected increase to 1.12 billion by 2030. At least 2.8 million people die each year as a result of obesity. In China, the incidence of obesity is also rapidly rising. The number of obese people in China has reached 90 million, making it the country with the largest obese population in the world [2]. The evidence indicates that an effective way to prevent obesity and the consequent associated pathologies is to maintain a healthy lifestyle that includes a varied and balanced diet [3]. In recent years, it has been found that some natural plant extracts have good therapeutic effects, multiple action targets, and fewer side effects while preventing and treating obesity [4]. To improve diet and health, it is of great practical significance to discover natural functional components with local characteristics that are derived from raw food materials and to investigate their intervention effects and mechanisms on high-calorie diet-induced obesity.

*Auricularia auricula-judae*, known as “black treasure”, is a kind of saprophytic fungus that grows on rotted wood in forests and consists of two parts, the mycelium and fruiting body [5]. *Auricularia auricula-judae* was first recorded in “Shen Nong’s Materia Medica”. China is the main *Auricularia auricula-judae* producer in the world, and its annual output accounts for more than 90% of the world’s total output. Northeast *Auricularia auricula-judae* is particularly well-known [6]. Except for the northeast region, the Qinba mountainous area in Shaanxi Province has dense forests, sufficient sunlight, a humid climate, and a large temperature difference between day and night in summer and autumn, which are suitable conditions for the growth of *Auricularia auricula-judae*. Polysaccharides are one of the most important active components in the fruiting body of *Auricularia auricula-judae*, which promotes antioxidant, immunomodulatory, antitumor, anticoagulant, antifatigue, and other activities [7,8]. Studies have found that *Auricularia auricula-judae* and its polysaccharides can reduce serum triglycerides and cholesterol [9,10,11], but the molecular mechanism of its influence on lipid metabolism is still unclear. At present, most research focuses on the well-known Northeast *Auricularia auricula-judae* resources. Due to the complex structure of polysaccharides and the difficulty in separation and purification, different extraction methods often affect the yield, structural characteristics, and biological activity of natural polysaccharides. Studies on *Auricularia auricula-judae* polysaccharides (AAP) in the Qinba Mountain area are rarely reported. Our research team previously found that the hot water-extracted *Auricularia auricula-judae* polysaccharides in the Qinba Mountains were mainly composed of mannose, glucuronic acid, and xylose, with a relative molar percentage of 60.87:20.83:9.86, which was similar to the polysaccharides that were extracted from *Auricularia auricula-judae* in northeast China. The free radical scavenging experiment results in vitro showed that both of them had good antioxidant capacities. However, it is not clear whether AAP from the Qinba Mountains has good weight loss and lipid-lowering potential, and its molecular mechanism that is related to metabolic disorders that are caused by a high-calorie diet needs to be further studied.

Therefore, this study aims to explore the intervention effect of AAP from the Qinba Mountains on obesity in mice that is induced by high-fat and high-fructose diets (HFFD) and to investigate the underlying molecular mechanism of the potential anti-obesity effect of AAP based on lipid homeostasis and inflammatory response regulation. This study is expected to provide a theoretical basis for guiding residents’ rational diet structure and solving the problem of overnutrition. It has positive significance for developing and utilising *Auricularia auricula-judae* resources in the Qinba Mountain area and nutritional and functional foods with *Auricularia auricula-judae* polysaccharides as the main functional component.

## 2. Materials and Methods

### 2.1. Samples Preparation

The fruiting bodies of *Auricularia auricula-judae* (purchased from Shaanxi Tianmei Green Industry Co., Ltd. (Hanzhong, China), and grown in the Qinba Mountain area of northwest China) were crushed with a multi-functional crusher (200y, Yongkang Boou Hardware Products Co., Ltd., Yongkang, China) and sieved by 40-mesh sieve. *Auricularia auricula-judae* powder (30 g) was extracted with boiling water (material–liquid ratio was 1:50) at 100 °C for 2 h. After centrifugation at 2808× *g* for 15 min, the supernatant was transferred into a fresh tube, and the precipitate was re-extracted. The supernatants were combined and concentrated using a rotary evaporator (R-1005, Zhengzhou Changcheng Science Technology and Trade Co., Ltd., Zhengzhou, China). Then, anhydrous ethanol was added into the concentrated solution in a ratio of 1:4 (volume fraction) for ethanol precipitation for 12 h. After centrifugation at 2808× *g* for 15 min, the supernatant was discarded, and the residue was retained and dissolved in hot water. After the residue was fully dissolved, the protein was removed by the Sevage method. Specifically, Sevage reagent (dichloromethane: n-butanol = 4:1, *v/v*) was added into the solution and stirred for 20 min to cause the protein to fully denature and precipitate out. After centrifugation at 2808× *g* for 15 min, the supernatant was transferred to an aqueous solution, and the precipitation layer and organic layer were discarded. The deproteinization procedure was repeated 4~5 times. The cellophane with a molecular cut-off of 3000 Da was used for dialysis in a flowing tap water for 3 days. Then, the solution was concentrated by a rotary evaporator, freeze-dried, and finally weighed and bagged to obtain a crude polysaccharide sample of *Auricularia auricula*-*judae.* The carbohydrate content, protein content, and uronic acid content of AAP were 53.15 ± 4.52%, 1.45 ± 0.12%, and 11.04 ± 0.59%, respectively. The AAP was mainly composed of mannose, glucuronic acid, xylose, and the relative mole percentage was 60.87:20.83:9.86.

### 2.2. Experimental Design

A total of 60 7-week-old SPF C57BL/6J male mice were purchased from the Experimental Animal Center of Xi’an Jiaotong University (license number: SCXK 2018-001). The animal experiment protocol was approved by the Animal Ethics Committee of the Laboratory Animal Center of Northwest University (Approval Code: NWU-AWC-20200401M) and was carried out in accordance with the “Administrative Regulations on Laboratory Animals” of the National Science and Technology Commission of the People’s Republic of China. The mice were reared under standard conditions (temperature 25 ± 2 °C, relative humidity 40 ± 10%, light-dark cycle of 12/12 h, clean bedding, and free water and food intake). Standard feed (AIN-93M) and 45% (TP230100) high-fat feed were purchased from TROPHIC Animal Feed High-Tech Co., Ltd. (Nantong, China) and stored in a refrigerator at 4 °C. The energy content of the standard diet was 3.6 kcal/g, and the energy content of the 45% high-fat diet was 4.5 kcal/g. After one week of adaptation, the mice were randomly divided into 6 groups, with 10 mice in each group. The specific treatment methods were as follows (Figure 1A). The mice in the normal diet (ND) group were fed with AIN93M standard feed and distilled water. The mice in the HFFD induction group were fed with 45% high-fat feed and 10% high fructose drinking water. The mice in the ND+AAP negative control group were given standard feed containing 200 mg/kg/day AAP and distilled water. The HFFD + AAP treated group mice were given 50, 100, or 200 mg/kg/day AAP in 45% high-fat feed, respectively, and 10% high-fructose drinking water. The experimental period was 12 weeks. All surgery was performed under anaesthesia, and all efforts were made to minimise suffering.

### 2.3. Assessments

#### 2.3.1. Body Weight, Food Intake, Fluid Intake Measurement

During the 12 weeks of rearing, the body weight, food intake, and drinking water intake of the mice were recorded every week. The average food intake, energy intake, and food efficiency ratio of the mice were calculated. Wherein the energy intake (kcal) = food intake (g) × feed energy content (kcal/g) + drinking water intake (mL) × fructose water energy content (kcal/mL); food efficiency ratio (FER) = Weight gain (g)/energy intake (kcal).

#### 2.3.2. Intraperitoneal Glucose Tolerance Test and the Homeostasis Model Assessment of Insulin Resistance Index

The mice were fasted one day before the end of the feeding cycle. Blood was taken from the mice’s tail vein, and the fasting blood glucose and fasting insulin levels of the mice were measured immediately by ELISA kits (Shanghai Xinle Biotechnology Co., Ltd., Shanghai, China). For the glucose tolerance test, glucose (2 g/kg body weight) was injected intraperitoneally, and the plasma glucose levels were measured before and 15, 30, 60, and 120 min after the injection. We used the following formula to calculate the homeostasis model assessment of insulin resistance (HOMA-IR) index:(fasting insulin concentration (mU/L) × fasting glucose concentration (mg/dL) × 0.05551)/22.5(1)

#### 2.3.3. Blood Sample Collection and Tissue Weight Measurement

The mice were fasted for 12 h and then weighed. After anaesthesia, blood samples were collected from the retro-orbital sinus/plexus. The plasma was centrifuged at 1096× *g* for 30 min at room temperature to obtain mice serum. The mice were sacrificed, and mouse liver, subcutaneous white adipose tissue (iWAT), and epididymal white adipose tissue (eWAT) were immediately separated, washed with PBS buffer, weighed, and quickly stored at −80 °C or fixed in 4% paraformaldehyde at 4 °C. We used the following formula to calculate the liver index (%):Liver index (%) = liver mass/body mass × 100%(2)

#### 2.3.4. Lipid Analysis

For the determination of the serum lipid levels, Enzymatic assay kits (Nanjing Jiancheng Bioengineering Institute, Nanjing, China) were used to measure the total serum cholesterol (TC, A111-1-1), triglyceride (TG, A110-1-1), low-density lipoprotein (LDL-C, A113-1-1), and high-density lipoprotein (HDL-C, A112-1-1) levels according to the kit instructions.

For the extraction and detection of serum metabolite, 50 μL of the serum sample was transferred to a centrifuge tube, and 200 μL of the extraction solution (methanol: acetonitrile = 1:1 (*v/v*), containing the isotope-labelled internal standard mixture) was added and vortexed for 30 s. Then, it was ultrasonicated for 10 min (ice-water bath) and incubated at 40 °C for 1 h. Then, the samples were centrifuged at 4 °C 10,010× *g* for 15 min. The supernatant was taken and tested on a computer in a sample inlet bottle. Another equal amount of supernatant was taken from all the samples and mixed into the QC samples for testing on the computer. The target compounds were separated on a Waters Acquity UPLC Behamide (2.1 mm × 100 mm, 1.7 μm) LC column using a Vanquish (Thermo Fisher, Shanghai, China) UPLC. Phase A of the liquid chromatography was an aqueous phase containing 25 mmol/L ammonium acetate and 25 mmol/L ammonia, and Phase B was acetonitrile. The temperature of the sample tray was 4 °C, and the injection volume was 2 μL. The Thermo Q Exactive HFX mass spectrometer is capable of primary and secondary mass spectral data acquisition under the control of control software (Xcalibur, Thermo Fisher, Shanghai, China). Detailed parameters are as follows: Sheath gas flow rate: 30 Arb; Aux gas flow rate: 25 Arb; Capillary temperature: 350 °C; Full ms resolution: 60,000; MS/MS resolution: 7500; Collision energy: 10/30/60 in NCE mode; Spray Voltage: 3.6 kV (positive) or −3.2 kV (negative). After the original data were converted into mzXML format using the ProteoWizard software, the independently written R program package (the kernel was XCMS) was used for peak recognition, peak extraction, peak alignment, and integration, and then, the data were matched with the two-stage mass spectrometric database that was built by BiotreeDB (V2.1) for material annotation. The cut-off value of algorithm scoring was set to 0.3.

#### 2.3.5. HE Staining of Liver Tissue

The liver tissues of mice were fixed in 4% paraformaldehyde and embedded in paraffin for HE staining. The tissue sections were placed in xylene I for 10 min, xylene II for 10 min, 100% ethanol I for 5 min, 100% ethanol II for 5 min, 90% ethanol for 5 min, 80% ethanol for 5 min, and 70% ethanol for 5 min, for dewaxing and rehydration. Then, the tissues were washed 3 times with PBS for 5 min each. The nuclei were stained with haematoxylin for 5 min and then differentiated with 1% hydrochloric acid alcohol, and rinsed indirectly with tap water for 15 min, followed by adding 1% dilute ammonia water. The tissue cell plasma was dyed with eosin staining solution for 3 min, then dehydrated with ethanol, and made transparent with xylene. Finally, the film was sealed with neutral resin and observed through an optical microscope.

#### 2.3.6. Gene Expressions Analysis

Total RNA was extracted from liver tissues with a Takara MiniBEST Universal RNA Extraction Kit (Takara MiniBEST Universal RNA Extraction Kit, Dalian, China), and its purity was tested with the Nano-200 nucleic acid quantifier. The RNA was stored at −80 °C prior to further analysis by microarray and real-time quantitative PCR (RT-qPCR). According to the instructions of the reverse transcription kit (Takara PrimeScript RT Master Mix, Dalian, China), 1 mg of RNA was reversed to cDNA with a 9600 gene thermal cycler PCR machine. To detect the target gene mRNA levels, real-time quantitative PCR was performed by the CFX96TM real-time system (Bio-Rad, Hercules, CA). The mouse-derived primers are shown in Table 1. *Gapdh* was used as an internal control, and 2^−ΔΔCt^ was used to calculate the relative gene expression.

#### 2.3.7. Statistical Analysis

Each experiment was repeated at least six times, and the obtained data were expressed as the mean ± standard error (X ± SEM). Significant differences between the mean values were determined by a one-way ANOVA using SPSS 19.0. The means were considered statistically significant if *p* < 0.05.

## 3. Results

### 3.1. Dietary Supplementation of AAP Reduced the Food Efficiency Ratio and Slowed Down the Bodyweight Gain of Obese Mice Fed by HFFD

Bodyweight gain is the most direct manifestation of obesity. As shown in Figure 1A, the mice in the HFFD group were significantly fatter than those in the ND group. However, when 100 mg/kg/day AAP was added, this phenomenon was obviously weakened. The bodyweight of the HFFD-fed mice increased remarkably compared with that of the ND group in the second week (*p* < 0.05). Significantly, the weights of the mice in the AAP intervention group were lower than those in the HFFD group in the period from the 4th to 12th week (*p* < 0.05/*p* < 0.01) (Figure 1B). In addition, Figure 1C illustrates that, at the end of feeding, the bodyweight gain of the mice in the HFFD group was also markedly higher than that in the ND group (*p* < 0.01), while AAP reversed this trend (*p* < 0.05/*p* < 0.01). The above results indicate that dietary supplementation of AAP significantly alleviated the weight gain of mice that was caused by HFFD.

The accumulation of subcutaneous fat and epididymal fat in mice is demonstrated in Figure 1C. Compared with the ND group, the weight of subcutaneous fat and epididymal fat in the HFFD treatment group was boosted obviously (*p* < 0.01), and the supplementation of 100 mg/kg/day and 200 mg/kg/day AAP significantly (*p* < 0.05) inhibited the accumulation of subcutaneous and epididymal adipose that was caused by HFFD.

To further explore whether the increase in the weight of mice was caused by the increase in food intake, the energy intake and FER were calculated during the experiment period. The results in Figure 2A show that the food intake of the HFFD-fed mice was markedly reduced, which may be due to the fact that HFFD was a diet rich in energy and mice required less food to maintain normal life activities. When compared with the ND group, a better flavour may contribute to the increased food intake after adding AAP. As for the total fluid intake (Figure 2B), there was no significant difference among all the groups. It was calculated that the energy intake and FER of the HFFD group were markedly higher than those of the ND group (Figure 2C,D), and dietary supplementation with low, medium, and high doses of AAP could effectively inhibit the increase in the FER value, whether fed with a normal diet or a high-fat diet (*p* < 0.01) (Figure 2D).

### 3.2. Dietary AAP Supplementation Improved Glucose Tolerance and Insulin Resistance in Obese Mice Induced by HFFD

Studies have shown that high-fat diet-induced obesity is usually accompanied by systemic insulin resistance [12]. In order to explore the regulatory effect of AAP on glucose tolerance, the fasting glucose levels and glucose tolerance test (GTT) were carried out after feeding for 12 weeks. Figure 3B illustrates that the fasting glucose level of mice in the HFFD induction group was remarkably higher than that in the ND group (*p* < 0.01), and supplementation of different doses of AAP obviously reversed this phenomenon (*p* < 0.05/*p* < 0.01). As shown in Figure 3A, which is the result of the GTT test, within 15 min after the intraperitoneal injection of 2 g/kg glucose, the blood glucose level of mice in the HFFD group was notably increased (*p* < 0.01). However, AAP alleviated the increase in blood glucose that was caused by the HFFD. After 30 min, the blood glucose levels of the mice in all the groups began to decrease. The blood glucose reduction in the HFFD + AAP group with different concentrations was significantly greater than in the HFFD-induced obesity mice.

Furthermore, the fasting serum insulin level is an important indicator of insulin resistance. After 12 weeks of feeding, the mice in the HFFD group developed hyperinsulinemia, and the fasting insulin values were strikingly boosted (*p* < 0.01), indicating that HFFD induced insulin resistance in mice. Interestingly, the dietary supplement of AAP remarkably improved this trend (*p* < 0.01) (Figure 3C). Similar modulatory effects were also illustrated by HOMA-IR analysis (Figure 3D).

### 3.3. Dietary Supplement of AAP Regulated Serum Lipid Levels in HFFD-Induced Obese Mice

In order to explore the intervention effect of AAP on the blood lipid metabolism disorder of obese mice, the levels of TG, TC, HDL-C, and LDL-C in the serum of mice were detected. As shown in Figure 4A,B, the dietary supplementation of AAP significantly suppressed the increase in TG and TC levels that were caused by the HFFD (*p* < 0.05/*p* < 0.01). Moreover, the findings in Figure 4C,D demonstrate that the HFFD markedly increased LDL-C levels (*p* < 0.01) and decreased HDL-C levels (*p* < 0.01), whereas the diets that were supplemented with 100 mg/kg/day or 200 mg/kg/day AAP significantly down-regulated LDL-C expression (*p* < 0.01), indicating that AAP had abilities to balance serum lipid metabolism disorder that was caused by the HFFD.

In addition, lipids and lipid-related metabolites in mouse serum were detected by mass spectrometry and the results are illustrated in Figure 5. In positive ion mode (Figure 5A), compared with obese mice that were induced by HFFD, the dietary supplementation of AAP significantly up-regulated the levels of serum lipid-related metabolites, such as L-Palmitoylcanitine, PC (16:0/16:0), Dodecanoylcarnitine, PC (P-16:0/16:0), PC (22:6(4Z,7Z,10Z,13Z,16Z,19Z)/18:0), L-Hexanoylcarnitine, PC (22:6(4Z,7Z,10Z,13Z,16Z,19Z)/20:1(11Z)), Linoleyl carnitine, PC (P-16:0/18:0), L-Octanoylcarnitine, LysoPE (22:6(4Z,7Z,10Z,13Z,16Z,19Z)/0:0), Elaidic carnitine, and trans-Hexadec-2-enoyl carnitine (*p* < 0.05), while it down-regulated Glycerophosphocholine, LysoPE (18:3(6Z,9Z,12Z)/0:0), PC (22:2(13Z,16Z)/14:0), LysoPE (18:1(9Z)/0:0), 7-Ketocholesterol, PC (18:2(9Z,12Z)/18:0), LysoPE (18:2(9Z,12Z)/0:0), Campesterol, PC (18:2(9Z,12Z)/15:0), LysoPC (16:1(9Z)/0:0), LysoPC (18:2(9Z,12Z)), N-Palmitoylsphingosine, LysoPC (22:4(7Z,10Z,13Z,16Z)), and Cer (d18:0/16:0) levels (*p* < 0.05). Then, the MetaboAnalyst database was used to analyze the metabolic pathways of these metabolites. It was found that these metabolites participated in glycerophospholipid metabolism, linolenic acid metabolism, alpha-Linolenic acid metabolism, ether lipid metabolism, arachidonic acid metabolism, fatty acid degradation, and steroid biosynthesis. In addition, in the negative ion mode (Figure 5B), the serum concentrations of 10 metabolites including Dodecanoic acid, Palmitoleic acid, 12-Methyltridecanoic acid, 2-Ethyl-2-Hydroxybutyric acid, Succinic acid semialdehyde, isoformous acid, Cortisone, Traumatic acid, LysoPA (16:0/0:0), and 15-Keto-prostaglandin E2 showed statistically significant differences (*p* < 0.05). These metabolites were associated with metabolic pathways such as fatty acid biosynthesis, butanoate metabolism, alanine, aspartate, and glutamate metabolism, biosynthesis of unsaturated fatty acids, and steroid hormone biosynthesis.

### 3.4. Dietary AAP Supplementation Reduced Mice Liver Weight Gain and Alleviated Liver Histomorphology Changes Induced by HFFD

The liver weight is a basic indicator reflecting the health status of the liver [13]. As illustrated in Figure 6A,B, the liver weight and liver index of the mice that were fed by HFFD were remarkedly higher than those in the ND group (*p* < 0.01). However, the low, medium, and high doses of AAP intervention groups significantly inhibited the increases in liver tissue weight and liver index that was caused by HFFD (*p* < 0.01). Haematoxylin and eosin (HE) staining was further performed to observe the tissue state of the liver. After HE staining, the nuclei and cytoplasm of the liver tissues were stained blue and red, respectively. Compared with the ND group, the liver in the HFFD model group showed steatosis, enlarged hepatocytes, and a large number of fat droplet vacuoles in the cytoplasm (Figure 6C). Obviously, the hepatomegaly of the liver in AAP intervention mice was lightened to some degree, and the vacuoles of fat droplets in the cytoplasm were decreased. It showed that AAP significantly alleviated the morphological changes in liver tissue that was induced by HFFD.

### 3.5. Dietary Supplement of AAP Inhibited Liver Inflammatory Response in HFFD-Fed Mice

Excessive accumulation of lipid droplets in the liver will increase the risk of hepatitis and nonalcoholic fatty liver disease [14]. As shown in Figure 7, when compared with ND mice, the levels of inflammatory mediators such as *IL-6*, *IL-1β,* and *TNF-α* mRNA in the liver of the HFFD model mice increased (*p* < 0.01), whereas the levels of anti-inflammatory mediators *IL-10* mRNA decreased markedly (*p* < 0.01), which was consistent with the reference. AAP significantly (*p* < 0.05/*p* < 0.01) inhibited the gene expressions of *IL-6*, *IL-1β,* and *TNF-α* and promoted the gene expressions of *IL-10*, thus suppressing the mice’s liver inflammation response that were fed by HFFD.

### 3.6. AAP Down-Regulated the Pro-Lipid Accumulation Genes Expressions, Up-Regulated Pro-Lipolysis Genes and Mitochondrial Active Genes Expressions, Improving Liver Lipid Metabolism

In order to further investigate the potential mechanism of AAP in inhibiting the excessive accumulation of liver lipid droplets that were caused by HFFD, the mRNA levels of genes that are related to lipogenesis regulation were detected in this study. As demonstrated in Figure 8A, the expressions of lipogenesis-related genes, including peroxisome proliferator-activated receptor γ (*Pparg*), cholesterol regulatory element-binding protein 1c (*Srebp1c*), acetyl-CoA carboxylase (*Acaca*), fatty acid synthase (*Fasn*), and malic enzyme (*ME1*), were significantly higher in the HFFD treatment group than in the ND group (*p* < 0.01). Compared with the HFFD group, the mRNA levels of the above genes were markedly decreased after supplementation with different doses of AAP, suggesting that AAP exerts good regulatory activity on mouse liver lipogenesis.

The liver cholesterol metabolism-related gene expressions are shown in Figure 8B. The expressions of cholesterol synthesis regulatory genes, including hydroxymethylglutaryl-CoA reductase (*Hmgcr*) and cholesterol acyltransferase 2 (*Acat2*) in the HFFD fed mice liver, were markedly boosted, while AAP significantly down-regulated *Hmgcr* and *Acat2* mRNA expressions (*p* < 0.05/*p* < 0.01). Besides, it has been reported that liver receptors (*LXR*), *ABCG5,* and *ABCG8* can promote the elimination of cholesterol from the body. After HFFD treatment, the expression of *LXRβ* in the mice livers decreased obviously, whereas the expressions of *LXRβ*, *Abcg5*, and *Abcg8* were enhanced in the HFFD+AAP group (*p* < 0.05/*p* < 0.01) (Figure 8B). These results demonstrated that AAP had a certain regulatory effect on cholesterol metabolism.

To further explore whether the specific mechanism of AAP in regulating lipid metabolism was related to the biological activity of mitochondria, the mRNA levels of mitochondrial-related lipid metabolism genes were detected. As shown in Figure 8C, HFFD significantly reduced the expressions of fatty acid β-oxidation-related genes *Cpt1α* and *Cpt2*, and AAP supplementation could notably reverse this tendency (*p* < 0.05/*p* < 0.01). In addition, genes that are related to mitochondrial function including *Pgc1α*, *Sirt1*, *COXII*, and *TFAM* showed the same trend as the above genes.

## 4. Discussion

The United Nations has proposed “one meat, one vegetable, and one mushroom” as a rational dietary structure in the 21st century. Each year, about six million tons of edible mushrooms are commercially produced around the world. Modern medical research has shown that edible fungi such as *Ganoderma lucidum*, shiitake mushrooms, and Tremella have immune regulating, antitumor, and hypoglycaemic effects [15,16,17,18]. Among them, the main active ingredient, edible mushroom polysaccharide, has exerted great advantages in repairing human cells, treating hepatitis, cardiovascular diseases, and anti-ageing, and has become an important pathway for humans to screen new pharmaceutical components [19,20]. The results of this study demonstrate that dietary supplementation with polysaccharides that were extracted from *Auricularia auricula-judae* significantly inhibited mouse subcutaneous and epididymal fat accumulation, decreased serum TG and TC levels, remarkably down-regulated LDL-C expression, and slowed down weight gain.

Eating habits are the main cause of obesity [21]. In addition to a high-fat diet, the influence of fructose on the occurrence and development of obesity and fatty liver is particularly noticeable. Fructose, an isomer of glucose, is the sweetest natural sweetener and is abundant in soft drinks and other foods. Since the introduction of high fructose corn syrup (HFCS) into the food processing industry in the 1970s, fructose intake has increased year by year, especially for some teenagers who drink soft drinks instead of water [22]. Fructose metabolism in the body is different from glucose metabolism in many ways [23,24]. Fructose promotes food intake and slows energy metabolism in a resting state. In addition, without increasing the energy intake, fructose could bypass the key rate-limiting step in the glycolytic pathway that is regulated by the cell energy state and generate excessive acetyl coenzyme A to enter the de novo fat synthesis pathway to synthesise fat [25]. The most important difference is that fructose, when metabolized intracellularly, causes rapid and irreversible ATP consumption, purine nucleotide conversion, and eventually, uric acid production. Fructose-induced uric acid production can reduce fatty acid oxidation, especially through inducing mitochondrial oxidative stress to activate the fat synthesis pathway, leading to obesity and visceral fat accumulation [26,27]. α-glucosidase is a key enzyme regulating glucose metabolism. Previous studies have shown that the α-amylase activity inhibition rate of 1 mg/mL AAP was 58.52%, which was obviously higher than that of *Auricularia cornea var. Li*, *Auricularia polytricha*, and Mulberry polysaccharides [28]. AAP and its in vitro hydrolysate could significantly improve the body weight and blood sugar changes of streptozotocin-induced type 2 diabetic rats, reduce TG and LDL-C levels, as well as promote insulin secretion and hepatic glycogen synthesis [29]. The results of this study found that the addition of AAP significantly inhibited the increase in the blood glucose and insulin levels and decreased HOMA-IR index, indicating that AAP had positive regulatory effects on insulin resistance (Figure 3).

Numerous studies have shown that natural polysaccharides can improve the disorder of lipid metabolism by changing genes that are related to lipid metabolism. Polysaccharides regulate TG metabolism mainly through three pathways, the ATGL-(PPAR-α)-(PGC-1α) pathway, the (SREBP-1c)-ACC/FAS pathway, and the ACC-CPT1 pathway, and regulate cholesterol metabolism primarily through the (SREBP-2)-HMGCR pathway, the PCSK9-LDLR pathway, and bile acid synthesis pathway [30]. Wu et al. screened out 13 genes that are involved in lipid metabolism in the liver and epididymal tissue by gene expression profiling array and found that black tea polysaccharides inhibited fat formation, accelerated fat digestion, and promoted lipolysis by regulating the expressions of differential genes affecting lipid metabolism, including bile acid secretion, transforming growth factor signalling, insulin signalling, glycolipid metabolism, fatty acid degradation, and the AMPK signalling pathway [31]. The polysaccharide from *Cyclocarya paliurus* leaves exerts therapeutic effects on hyperlipidaemic rats through the induction of ATGL and PPAR-α and the down-regulation of FAS and HMGCR [32]. Schisandra polysaccharide has been shown to markedly suppress the hepatic lipogenesis related genes, SREBP-1c, ACC, and FAS expressions [33]. The intervention of 800 mg/kg/d green tea polysaccharides could up-regulate the expressions of CPT-1, down-regulate the expressions of PPARγ, SREBP-1c, FAS, and LXRα, thus significantly reducing the fat index and adipocyte area, and inhibiting mouse obesity [34,35]. In this study, we found that AAP down-regulated the mRNA expressions of liver adipogenic genes *Pparg*, *Srebp1c*, *Fasn*, *Acaca,* and *ME1*; suppressed cholesterol synthesis-related genes *Hmgcr* and *Acat2* mRNA levels; up-regulated fatty acid β-oxidation-related genes *Cpt1α* and *Cpt2* expressions; and promoted the expressions of cholesterol efflux-related genes *LXR*, *Abcg5*, and *Abcg8*, thus improving the homeostasis of liver lipid metabolism in mice (Figure 8). Furthermore, it has been demonstrated that PGC-1α plays an important role in a variety of metabolic processes, including mitochondrial biogenesis and mitochondrial β-oxidation, and PGC-1α has emerged as an important therapeutic target for fatty liver disease [36]. AAP up-regulated *Pgc1α* and *Sirt1* expressions, and its intervention on HFFD-induced liver lipid metabolism disorder was closely related to the regulatory effects on mitochondrial function. However, as a non-digestible polysaccharide, how AAP acts on the target tissues and exerts its anti-obesity effect needs further in-depth study.

The body is in a state of chronic inflammation when obesity occurs, and the levels of TNF-α, ILs, and other inflammatory factors in the circulation and tissues of obese patients are significantly increased [37]. Fat accumulation in obesity leads to the hypertrophy of adipocytes, putting cells in a state of stress and activating pro-inflammatory pathways. In addition, the excessive enlargement of adipocytes leads to increased lipolysis, and the formation of metabolic endotoxemia (increased lipopolysaccharide and endotoxin in circulation) also contributes to inflammation [38,39]. The inflammatory factor TNF-α induces the decomposition of adipocytes to release free fatty acids, which, in turn, bind to Toll-like receptors on the surfaces of macrophages and adipocytes, further activating inflammatory pathway signals and promoting inflammatory factor release [40]. Furthermore, the transduction of insulin signals may be interfered with by inflammatory factors, resulting in insulin resistance, which then causes various metabolic disorders [41]. Polysaccharides have attracted the increasing attention of scientists around the world for their safety, potent anti-inflammatory, and immunomodulatory properties [42]. In our study, 50, 100, and 250 µg/mL of *Polygonatum sibiricum* polysaccharides alleviated IL-1β, IL-6, and TNF-α levels and promoted proliferation, glucose uptake, and glucose transporter 4 expression in IR-3T3-L1 adipocytes [43]. Wu et al. reported that Ganoderma lucidum beta 1,3/1,6 glucan (MBG) suppressed high-cholesterol diet-induced inflammation in male C57BL/6J mice by stimulating serum IgA and IgG production, boosting poly-Ig receptor expression, and increasing IL-2 production by NK cells [44]. The results of our study demonstrate that AAP markedly down-regulated the gene expressions of *IL-6*, *IL-1β,* and *Tnf-α*, while up-regulating the gene expression of anti-inflammatory factor *IL-10* in liver tissue (Figure 7), which could be one of the mechanisms underlying its liver-protecting and weight-reducing effects.

In recent years, studies have shown that the multiple biological activities of polysaccharides may be closely related to the regulation of intestinal flora. On the one hand, intestinal flora can degrade the polysaccharides to generate monosaccharides, thereby promoting the absorption and utilization of the polysaccharides by the body, simultaneously generating metabolites with certain biological activity, such as short-chain fatty acids (SCFAs). On the other hand, the polysaccharides can directly increase the abundance of beneficial bacteria and reduce the abundance of harmful bacteria, improving the physiological levels of the body [45]. Zhao et al. compared the effects of six edible fungi (*Auricular*, *Flammulina Velutipes*, *Lentinus edodes*, *Agaricus bisporus*, *Pleurotus ostertaus*, and *Pleurotus eryngii*) on the composition and diversity of intestinal flora using an in vitro fermentation system. The results showed that edible fungi increased the content of SCFAs, thus reducing the pH of the fermentation broth, and had a significant effect on the diversity of intestinal flora. In particular, AAP could significantly increase the abundance of *Bifidobacterium* and *Bacteroides* and reduce the abundance of *Fusobacterium* [46]. Zhang et al. found that *Auricularia auricula-judae* powder and *Auricularia auricula-judae* crude polysaccharides both markedly reduced the levels of TC and LDL-C in model SD rats by regulating intestinal flora and its metabolites. Among them, *Auricularia auricula-judae* powder increased the abundance of *Prevost’s* bacteria and high-abundance SCFAs-producing bacteroides related to a diet that was rich in dietary fibre, while *Auricularia auricula-judae* polysaccharides mainly promoted the abundance of low-abundance SCFAs -producing bacteria, such as *Flavobacterium* and *Clostridium IV* [11]. Although it has been demonstrated that many natural polysaccharides can improve the intestinal flora diversity of mice that is induced by a high-calorie diet, it is difficult to prove the causal relationships based on correlation analysis. The effective dose, molecular mechanisms, and metabolic process of polysaccharides in regulating obesity need to be systematically investigated.

The structure of an active substance determines its function. The types and contents of the active components in crops from different origins and varieties are different. Additionally, different extraction methods often affect the yield, structural properties, and biological activity of natural polysaccharides. At present, the research on the lipid-lowering of AAP mainly focuses on the well-known Northeast *Auricularia auricula-judae* resources. Our research team previously compared the physicochemical properties and antioxidant activities of AAP in the Northwest Qinba Mountains with AAP in Heilongjiang and analyzed the differences between AAP that was obtained by different extraction methods, such as hot water extraction, enzymatic extraction, and ultrasonic extraction. The results showed that the AAP from the Qinba Mountains was mainly composed of mannose, glucuronic acid, and xylose, and the relative molar percentage was 60.87:20.83:9.86, which was similar to the northeast. In addition, the content of mannose and glucuronic acid in AAP that was prepared by hot water extraction was the highest. Mannose is not only an important monosaccharide for protein glycosylation in mammals but also an inefficient source of cellular energy. Sharma et al. found that a certain amount of mannose supplementation inhibited the weight gain of mice that were fed with a high-fat diet, as well as reduced the proportion of fat, prevented fatty liver, enhanced endurance and maximum oxygen consumption, and improved glucose tolerance [47]. Mannose increased the ratio of firmicutes to bacteroidetes in the gut microbiota of mice. Moreover, functional transcriptomic analysis of the mouse caecal microbiota revealed that the coherent changes in microbial energy metabolism were caused by mannose, and it is speculated that mannose played a role by reducing the energy that is produced by the metabolism of complex carbohydrates by intestinal flora and reducing energy intake [47]. The results of this study showed that dietary supplementation of 100 mg/kg·day of *Auricularia auricula-judae* water-extracted polysaccharide from the Qinba Mountains significantly reduced the food efficiency ratio, inhibited the accumulation of subcutaneous fat and epididymal fat, improved lipid metabolism, and slowed down the mice body weight gain that was induced by HFFD (Figure 1). Due to the complexity and diversity of polysaccharides structure, the relationship between molecular weight, monosaccharide composition, glycosidic bond connection mode, the advanced structure of polysaccharides, and their functional activities is not clear enough, the main active groups of AAP which play a role in weight loss need further in-depth investigation.

## 5. Conclusions

This study showed that AAP that was extracted from Qinba Mountain *Auricularia auricula-judae* exerted good abilities to inhibit nutritional obesity in mice, which was related to the regulations of the inflammatory response and lipid metabolism. Dietary supplementation of AAP significantly reduced the accumulation of subcutaneous and epididymal fat in mice, altered serum lipid metabolites, and slowed down body weight gain that was caused by HFFD. AAP remarkably suppressed fasting glucose and fasting insulin levels and improved the insulin resistance state in mice. Additionally, AAP supplementation decreased inflammatory factor expression and alleviated liver histomorphology changes that were caused by HFFD. Furthermore, AAP down-regulated the expressions of liver pro-lipid accumulation genes and up-regulated pro-lipolysis and mitochondrial active gene expressions, improving liver lipid metabolism. These results lay a theoretical basis for an in-depth study on the anti-obesity function of AAP and provide a scientific basis for the further development and utilization of *Auricularia auricula-judae* resources in the Qinba Mountain area.

## Figures and Tables

**Figure 1 foods-11-00942-f001:**
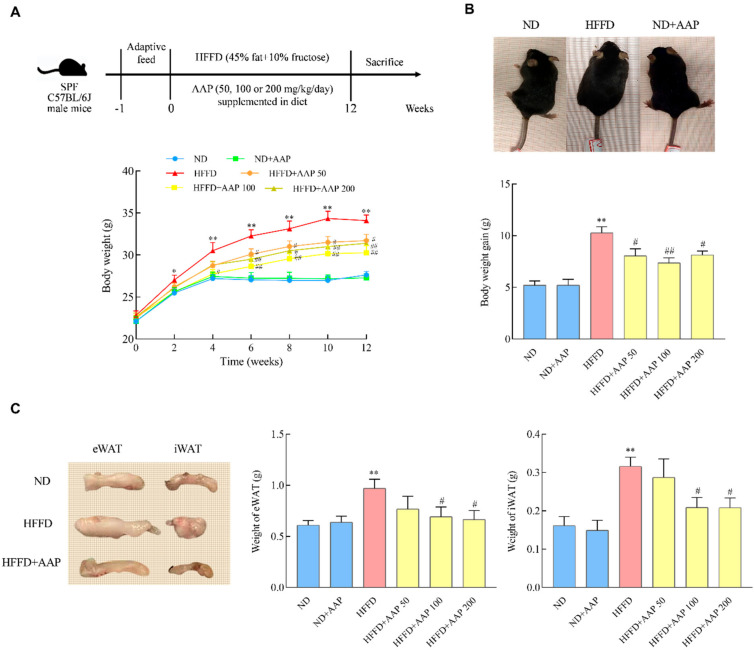
The effects of *Auricularia auricula**-judae* polysaccharides on HFFD induced mice body weight and fat accumulation. Seven-week-old male C57BL/6J mice were randomized into 6 groups (*n* = 10 per group): ND, ND + AAP, HFFD, and HFFD + AAP (*Auricularia auricula**-judae* polysaccharide, 50, 100, 200 mg/kg/day, in feed) for 12 weeks. (**A**) Above: Timeline of C57BL/6J mice with AAP and/or HFFD treatment; Below: Body weight; (**B**) Above: Representative mouse pictures after 12 weeks feeding; Below: Body weight gain; (**C**) Weight of subcutaneous fat and epididymal fat. The data are presented as the mean ± SEM, * *p* < 0.05, ** *p* < 0.01, vs. ND group, # *p* < 0.05, ## *p* < 0.01 vs. HFFD group.

**Figure 2 foods-11-00942-f002:**
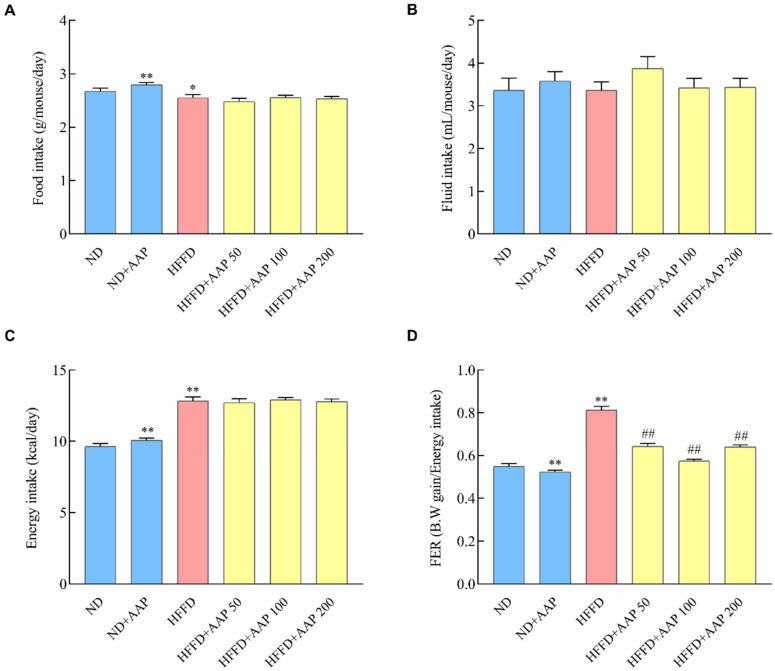
Effects of *Auricularia auricula-judae* polysaccharides on food intake and food efficiency ratio of mice. During the 12 weeks of mice rearing, the (**A**) food intake and (**B**) fluid intake of the mice were recorded every week. The calculated (**C**) energy intake and (**D**) food efficiency ratio of the mice. The data are presented as the mean ± SEM, * *p* < 0.05, ** *p* < 0.01, vs. ND group, ## *p* < 0.01 vs. HFFD group.

**Figure 3 foods-11-00942-f003:**
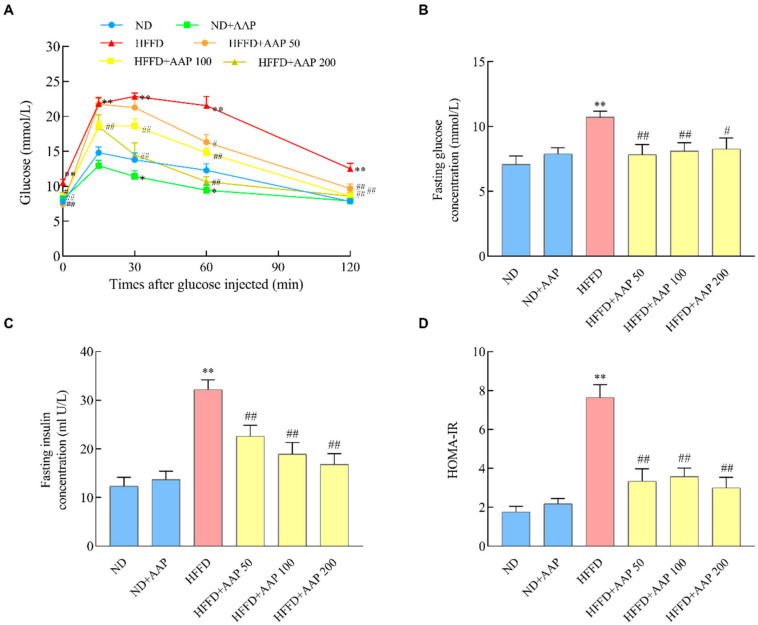
Effects of *Auricularia auricula-judae* polysaccharides on glucose tolerance and insulin resistance in obese mice that was induced by HFFD. At the end of 12-week feeding, mice were detected by (**A**) glucose tolerance tests; (**B**) fasting glucose levels; (**C**) fasting insulin levels; (**D**) insulin resistance index, HOMA-IR. The data are presented as the mean ± SEM, *n* ≥ 6, * *p* < 0.05, ** *p* < 0.01, vs. ND group, # *p* < 0.05, ## *p* < 0.01 vs. HFFD group.

**Figure 4 foods-11-00942-f004:**
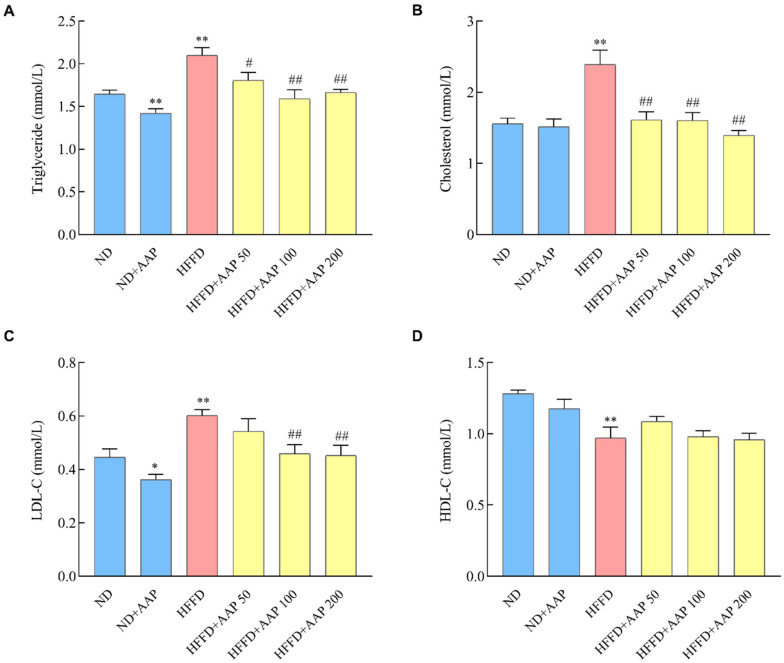
Effects of *Auricularia auricula-judae* polysaccharides on serum lipid levels in the HFFD-induced obese mice. (**A**) Triglyceride levels; (**B**) total cholesterol levels; (**C**) high-density lipoprotein cholesterol levels; (**D**) low-density lipoprotein cholesterol levels. The data are presented as the mean ± SEM, *n* ≥ 6, * *p* < 0.05, ** *p* < 0.01, vs. ND group, # *p* < 0.05, ## *p* < 0.01 vs. HFFD group.

**Figure 5 foods-11-00942-f005:**
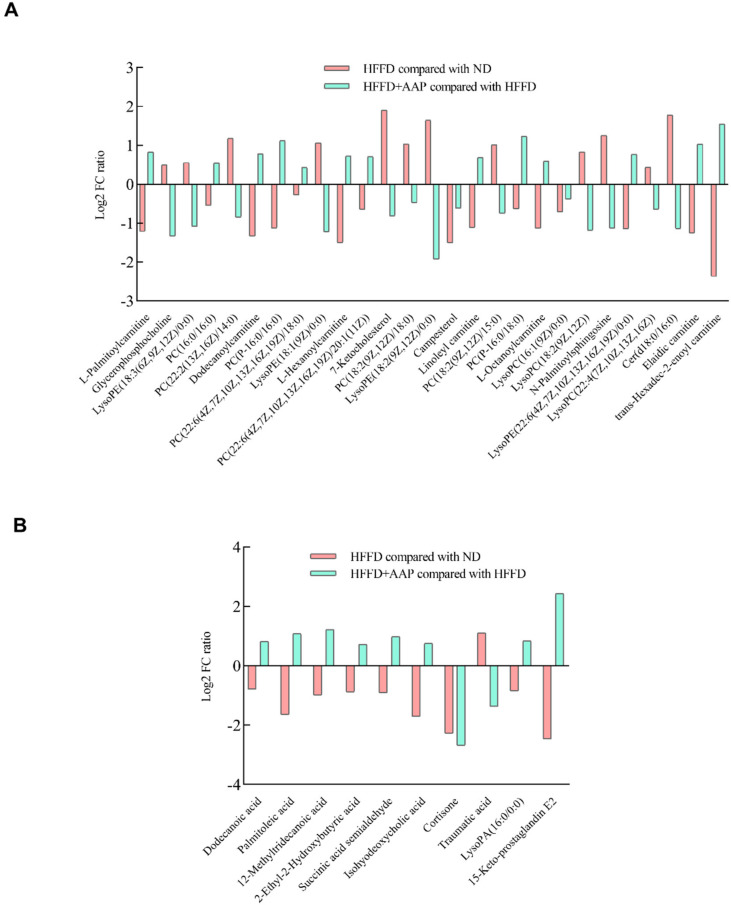
Effects of *Auricularia auricula-judae* polysaccharides on lipids and lipid-related metabolites in obese mouse serum. (**A**) Positive ion mode; (**B**) Negative ion mode. *n =* 6.

**Figure 6 foods-11-00942-f006:**
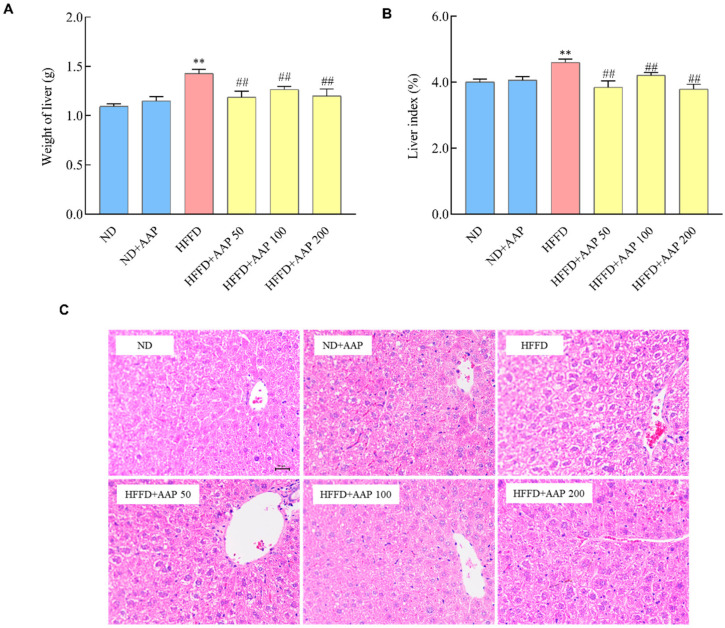
Effects of *Auricularia auricula-judae* polysaccharides on liver histomorphology changes in obese mice that were fed by HFFD. (**A**) liver weight; (**B**) liver index; (**C**) representative HE staining of liver (×400). The data are presented as the mean ± SEM, *n* ≥ 6, ** *p* < 0.01, vs. ND group, ## *p* < 0.01 vs. HFFD group.

**Figure 7 foods-11-00942-f007:**
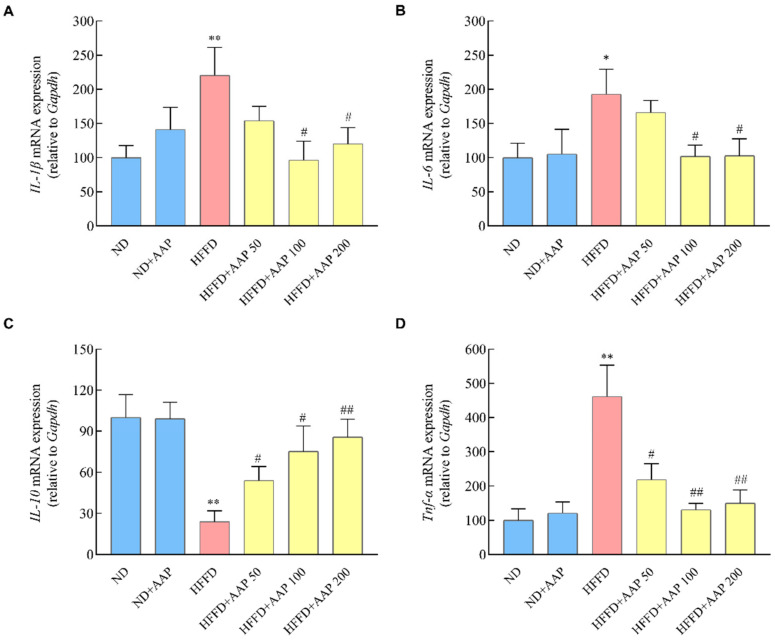
Effects of *Auricularia auricula-judae* polysaccharides on liver inflammatory response in HFFD-induced obese mice. The mRNA levels of inflammatory response including (**A**) *IL-1β*, (**B**) *IL-6*, (**C**) *IL-10*, (**D**) *TNF-α*. The data are presented as the mean ± SEM, *n* ≥ 6, * *p* < 0.05, ** *p* < 0.01, vs. ND group, # *p* < 0.05, ## *p* < 0.01 vs. HFFD group.

**Figure 8 foods-11-00942-f008:**
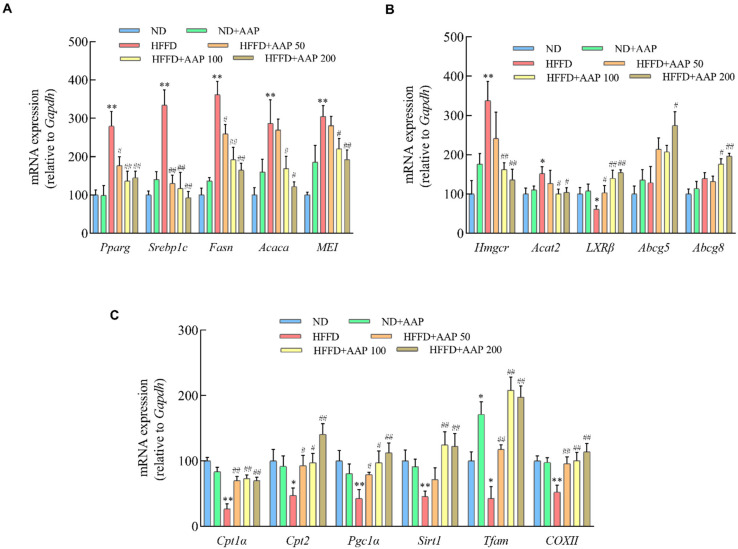
The effects of *Auricularia auricula-judae* polysaccharides on liver lipid metabolism-related gene expressions in obese mice that were induced by HFFD. (**A**) mRNA levels of lipogenic genes including *Pparg*, *Srebp1c*, *Fasn*, *Acaca*, and *ME1*; (**B**) The mRNA levels of cholesterol synthesis genes including *Hmgcr*, *Acat2*, *LXRβ*, *Abcg5*, and *Abcg8*; (**C**) mRNA levels of lipolytic genes and mitochondrial-related genes including *Cpt1a*, *Cpt2*, *Pgc1α*, *Sirt1*, *Tfam*, and *COXII*. The data are presented as the mean ± SEM, *n* ≥ 6, * *p* < 0.05, ** *p* < 0.01, vs. ND group, # *p* < 0.05, ## *p* < 0.01 vs. HFFD group.

**Table 1 foods-11-00942-t001:** Primer sequences that were used for semi-quantitative RT-PCR analysis.

Forward Primer		Reverse Primer
*Pparg*	TGCTGTTATGGGTGAAACTCTG	CTGTGTCAACCATGGTAATTTCTT
*Pgc1a*	GAAAGGGCCAAACAGAGAGA	GTAAATCACACGGCGCTCTT
*Fasn*	AGGTGGTGATAGCCGGTATGT	TGGGTAATCCATAGAGCCCAG
*Acaca*	CCGATTCATAATTGGGTCTGTGT	CCATCCTGTAAGCCAGAGATCC
*Cpt1a*	CGGTTCAAGAATGGCATCATC	TCACACCCACCACCACGATA
*Cpt2*	AGCCAGTTCAGGAAGACAGA	GACAGAGTCTCGAGCAGTTA
*LXRβ*	CGACTCCAGGACAAGAAGC	TCAAGAAGACACCACCAAGG
*Srebp1c*	AAGCAAATCACTGAAGGACCTGG	AAAGACAAGGGGCTACTCTGGGAG
*ME1*	AGTATCCATGACAAAGGGCAC	ATCCCATTACAGCCAAGGTC
*Acat2*	TTTGCTCTATGCCTGCTTCA	CCATGAAGAGAAAGGTCCACA
*Abcg5*	TGGCCCTGCTCAGCATCT	ATTTTTAAAGGAATGGGCATCTCTT
*Abcg8*	CCGTCGTCAGATTTCCAATGA	GGCTTCCGACCCATGAATG
*Sirt1*	TCTGTCTCCTGTGGGATTCC	GATGCTGTTGCAAAGGAACC
*Tfam*	GCTTCCAGGGGGCTAAGGAT	CCCAATCCCAATGACAACTC
*COX II*	GCCGACTAAATCAAGCAACA	CAATGGGCATAAAGCTATGG
*Hmgcr*	ATCCAGGAGCGAACCAAGAGAG	CAGAAGCCCCAAGCACAAAC
*IL6*	TTCCATCCAGTTGCCTTCTTG	TATCCTCTGTGAAGTCTCCTCTC
*IL10*	GCTCCAAGACCAAGGTGTCTACAA	CCGTTAGCTAAGATCCCTGGATCA
*IL1β*	TGACGGACCCCAAAAGATGA	TCTCCACAGCCACAATGAGT
*Tnfα*	CCCTCACACTCAGATCATCTTCT	GCTACGACGTGGGCTACAG
*Gapdh*	TGGAGAAACCTGCCAAGTATGA	TGGAAGAATGGGAGTTGCTGT

## Data Availability

The data presented in the manuscript is available on request from the corresponding author.

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
