# Peer review of "Dietary Supplementation of Auricularia auricula-judae Polysaccharides Alleviate Nutritional Obesity in Mice via Regulating Inflammatory Response and Lipid Metabolism"

_foods, 2022, doi:10.3390/foods11070942_

Round 1

Reviewer 1 Report

In general, the work is well structured and well written. Generally, I consider that the present work has good points such as: well-selected assays, the number of animals used is adequate and the results are quite goods. However I have some serious problems with different aspects of the job.  

Introduction: 

-I do not  understand what is intended to indicate in lines 32-33 with "western foods with high sugar content". Is it only the sugary food from that region  the problem? I do not see the punctuation necessary (please, remove it) 

-Lines 33-34 indicate that various pathologies are a global health problem, however the citations indicated (1 and 2) are studies carried out on rats. Please change to others of an epidemiological nature that support your statement 

-I do not completely agree with the authors' approach. In lines 40-43, you give the impression that “diet control” is not a way to deal with obesity or that have problems. The evidence indicates that the way to prevent obesity and the consequent associated pathologies is to maintain a healthy lifestyle that includes a varied and balanced diet. It seems to me that you suggest the use of polysaccharides from Auricularia auricula could be used for people who have a hypercaloric diet. Bearing in mind that the idea of the authors is the use of these polysaccharides for the design of functional foods or as a bioactive ingredient, they must keep in mind the preventive (not curative) nature of the food or its ingredients. Please rewrite the paragraph from lines 40-48 with the ideas above. 

Note: I saw a lack of consiscenty in the use of "Auricularia auricula" italics  Please cheack it.

Material and methods 

-Section 2.1. Given the complexity of the sample you use and the lack of characterization, you should go into more detail about sample preparation. If you have a reference where it is detailed, you must add it in this section. For example: “supernatant was concentrated” (line 91) how it was concentrated? Heating? Lyophilization? The centrifugation units should be in “G”.

-Review and homogenize the centrifugation units throughout the section.

-Section 2.6. What is the commercial house of the indicated kits? 

-Section 2.7 What does the acronym EP on line 150 mean? If it is only used there, it is better to put the full name.

-Section 2.8. In line 173 “w/s” is in italics. remove it. Line 176 “these” is capitalized (remove) 

Results and Discussion

Results are well described

The results are well described and correspond to what is indicated in the graphs.

Regarding the discussion, I consider that the role of the different disease markers, genes and metabolic enzymes is over-described. In addition, there are sections that are completely unnecessary (Lines 407-415), since they are described in the introduction. But the weakest part of the discussion is the lack of a clear hypothesis. I have not been able to understand the mechanism by which the polysaccharides in the sample are capable of performing their function. You have mentioned the microbiota, insulin... but the mechanism is not clear at all. You should support your hypothesis on other authors with similar compounds. Maybe that will help you understand what is going on. T

herefore, the discussion part needs greater depth and better contrast. 

Author Response

Reviewer 1:

In general, the work is well structured and well written. Generally, I consider that the present work has good points such as: well-selected assays, the number of animals used is adequate and the results are quite goods. However, I have some serious problems with different aspects of the job.

Responses:

Thank you for your encouraging comments. We have studied comments carefully and revised our manuscript seriously. The manuscript has been polished by MDPI language editing services (https://www.mdpi.com/authors/english), and the revised portions are highlighted in blue in the revised manuscript. We hope that the corrections will meet with approval.

Comments:

  1. Introduction: -I do not understand what is intended to indicate in lines 32-33 with "western foods with high sugar content". Is it only the sugary food from that region the problem? I do not see the punctuation necessary (please, remove it).

Responses:

Thanks for the reviewer’s kind remind. We have deleted the inappropriate descriptions in the revised manuscript.

Comments:

  1. Introduction: -Lines 33-34 indicate that various pathologies are a global health problem, however the citations indicated (1 and 2) are studies carried out on rats. Please change to others of an epidemiological nature that support your statement.

Responses:

Thanks for the reviewer’s kind remind. We have replaced the references [1] (Bluher, M., Obesity: global epidemiology and pathogenesis. Nature reviews. Endocrinology 2019, 15 (5), 288-298.) in the revised manuscript.

Comments:

  1. Introduction: -I do not completely agree with the authors' approach. In lines 40-43, you give the impression that “diet control” is not a way to deal with obesity or that have problems. The evidence indicates that the way to prevent obesity and the consequent associated pathologies is to maintain a healthy lifestyle that includes a varied and balanced diet. It seems to me that you suggest the use of polysaccharides from Auricularia auricula could be used for people who have a hypercaloric diet. Bearing in mind that the idea of the authors is the use of these polysaccharides for the design of functional foods or as a bioactive ingredient, they must keep in mind the preventive (not curative) nature of the food or its ingredients. Please rewrite the paragraph from lines 40-48 with the ideas above.

Responses:

Thanks for the reviewer’s valuable advice. The evidence indicates that an effective way to prevent obesity and the consequent associated pathologies is to maintain a healthy lifestyle that includes a varied and balanced diet. We have re-written the paragraph and added related reference [3] (Hwalla, N.; Jaafar, Z., Dietary Management of Obesity: A Review of the Evidence. Diagnostics 2020, 11 (1). 24.) in the revised manuscript. In addition, we have added some descriptions about natural plant extracts which have potent anti-obesity activities in the "Introduction" section, please see in Page 3, Line 45-49.

Comments:

  1. Introduction: Note: I saw a lack of consiscenty in the use of "Auricularia auricula" italics. Please check it.

Responses:

Thanks for the reviewer’s kind remind. We have checked and unified the writing format of “Auricularia auricula-judae” in the revised manuscript.

Comments:

  1. Material and methods -Section 2.1. Given the complexity of the sample you use and the lack of characterization, you should go into more detail about sample preparation. If you have a reference where it is detailed, you must add it in this section. For example: “supernatant was concentrated” (line 91) how it was concentrated? Heating? Lyophilization? The centrifugation units should be in “G”.

Responses:

Thanks for the reviewer’s kind remind. The extraction and physicochemical properties of Auricularia auricula-judae polysaccharides (AAP) were characterized previously (LIU Qian, ZHANG Xinyu, WANG Yan, WANG Anyu, ZHAO Yi, LONG Zixian, LIU Sining, DIE Yun, HUANG Linjuan. Physicochemical properties and antioxidant activities in vitro of Auricularia auricula polysaccharides in Qinba mountain area,2021,47(23):91- 97.DOI:10.13995/j.cnki.11-1802/ts.027141.). Since it is a Chinese journal, it cannot be cited in this study. We have re-written the extraction procedure of AAP in detail, and homogenized centrifugation units in “G”, please see in Page 4-5, Line 92-116 in the revised manuscript.

【The fruiting bodies of Auricularia auricula-judae (purchased from Shaanxi Tianmei Green Industry Co., Ltd., and grown in the Qinba Mountain area of northwest China) were crushed with a multi-functional crusher (200y, Yongkang BOU Hardware Products Co., Ltd.) and sieved by 40-mesh sieve. Auricularia auricula-judae powder (30 g) was extracted with boiling water (material–liquid ratio was 1:50) at 100 °C for 2 h. After centrifugation at 2808×g for 15 min, the supernatant was transferred into a fresh tube, and the precipitate was re-extracted. The supernatants were combined and concentrated using a rotary evaporator (R-1005, Zhengzhou Changcheng Science Technology and Trade Co., Ltd.). Then, anhydrous ethanol was added into the concentrated solution in a ratio of 1:4 (volume fraction) for ethanol precipitation for 12 h. After centrifugation at 2808×g for 15 min, the supernatant was discarded, and the residue was retained and dissolved in hot water. After the residue was fully dissolved, the protein was removed by the Sevage method. Specifically, Sevage reagent (dichloromethane: n-butanol =4:1, v/v) was added into the solution and stirred for 20 min to cause the protein to fully denature and precipitate out. After centrifugation at 2808×g for 15 min, the supernatant was transferred to an aqueous solution, and the precipitation layer and organic layer were discarded. The deproteinisation procedure was repeated 4~5 times. The cellophane with a molecular cut-off of 3000 Da was used for dialysis in flowing tap water for 3 days. Then, the solution was concentrated by a rotary evaporator, freeze-dried, and finally weighed and bagged to obtain a crude polysaccharide sample of Auricularia auricula-judae. The carbohydrate content, protein content, and uronic acid content of AAP were 53.15±4.52 %, 1.45±0.12 %, and 11.04±0.59 %, respectively. AAP was mainly composed of mannose, glucuronic acid, xylose, and the relative mole percentage was 60.87: 20.83: 9.86.】

Comments:

  1. Material and methods --Review and homogenize the centrifugation units throughout the section.

Responses:

Thanks for the reviewer’s kind remind. We have checked and homogenized the centrifugation units throughout “Material and methods” section in the revised manuscript.

Comments:

  1. Material and methods --Section 2.6. What is the commercial house of the indicated kits?

Responses:

Thanks for the reviewer’s kind remind. We have added the kits manufacturer information (Nanjing Jiancheng Bioengineering Institute, Nanjing, China) in the revised manuscript, please see in Page 7, Line 168-170.

Comments:

  1. Material and methods --Section 2.7 What does the acronym EP on line 150 mean? If it is only used there, it is better to put the full name.

Responses:

Thanks for the reviewer’s kind remind. We have revised the “EP tube” as “centrifuge tube” in the revised manuscript, please see in Page 7, Line 174.

Comments:

  1. Material and methods --Section 2.8. In line 173 “w/s” is in italics. remove it. Line 176 “these” is capitalized (remove)

Responses:

Thanks for the reviewer’s kind remind. We have deleted the “w/v” in line 173 and “these” in Line 176.

Comments:

  1. Results and Discussion—

Results are well described

The results are well described and correspond to what is indicated in the graphs.

Regarding the discussion, I consider that the role of the different disease markers, genes and metabolic enzymes is over-described. In addition, there are sections that are completely unnecessary (Lines 407-415), since they are described in the introduction. But the weakest part of the discussion is the lack of a clear hypothesis. I have not been able to understand the mechanism by which the polysaccharides in the sample are capable of performing their function. You have mentioned the microbiota, insulin... but the mechanism is not clear at all. You should support your hypothesis on other authors with similar compounds. Maybe that will help you understand what is going on. Therefore, the discussion part needs greater depth and better contrast.

Responses:

Thanks for the reviewer’s kind remind. The comment is very helpful to improve the quality of discussion part. We have revised the discussion parts, and deleted some unnecessary and inappropriate extended descriptions, please see in Page 22-28 in the revised manuscript.

We discussed the mechanism of polysaccharide's functional activity from the aspects of genes related to lipid metabolism, inflammation, intestinal flora, etc. In the discussion, we also compared with other polysaccharides, specifically as follows:

【Numerous studies have shown that natural polysaccharides can improve the disorder of lipid metabolism by changing genes related to lipid metabolism. Polysaccharides regulate TG metabolism mainly through three pathways, the ATGL-(PPAR-α)-(PGC-1α) pathway, the (SREBP-1c)-ACC/FAS pathway, and the ACC-CPT1 pathway, and regulate cholesterol metabolism primarily through the (SREBP-2)-HMGCR pathway, the PCSK9-LDLR pathway, and bile acid synthesis pathway [30]. Wu et al. screened out 13 genes involved in lipid metabolism in liver and epididymal tissue by gene expression profiling array and found that black tea polysaccharides inhibited fat formation, accelerated fat digestion, and promoted lipolysis by regulating the expressions of differential genes affecting lipid metabolism, including bile acid secretion, transforming growth factor signalling, insulin signalling, glycolipid metabolism, fatty acid degradation, and the AMPK signalling pathway [31]. The polysaccharide from Cyclocarya paliurus leaves exerts therapeutic effects on hyperlipidaemic rats through the induction of ATGL and PPAR-α and down-regulation of FAS and HMGCR [32]. Schisandra polysaccharide has been shown to markedly suppress the hepatic lipogenesis related genes, SREBP-1c, ACC, and FAS expressions [33]. The intervention of 800 mg/kg/d green tea polysaccharides could up-regulate the expressions of CPT-1, down-regulate the expressions of PPARγ, SREBP-1c, FAS, and LXRα, thus significantly reducing the fat index and adipocyte area, and inhibiting mice obesity [34,35]. In this study, we found that AAP down-regulated the mRNA expressions of liver adipogenic genes Pparg, Srebp1c, Fasn, Acaca, and ME1, suppressed cholesterol synthesis related genes Hmgcr and Acat2 mRNA levels, up-regulated fatty acid β-oxidation related genes Cpt1α and Cpt2 expressions, and promoted the expressions of cholesterol efflux related genes LXR, Abcg5, Abcg8, thus improving the homeostasis of liver lipid metabolism in mice (Figure 8). Furthermore, it has been demonstrated that PGC- 1α plays an important role in a variety of metabolic processes, including mitochondrial biogenesis and mitochondrial β-oxidation, and PGC-1α has emerged as an important therapeutic target for fatty liver disease [36]. AAP up-regulated Pgc1α and Sirt1 expressions, and its intervention on HFFD-induced liver lipid metabolism disorder was closely related to the regulatory effects on mitochondrial function. However, as a non-digestible polysaccharide, how AAP acts on the target tissues and exerts its anti-obesity effect needs further in-depth study.

The body is in a state of chronic inflammation when obesity occurs, and the levels of TNF-α, ILs, and other inflammatory factors in the circulation and tissues of obese patients are significantly increased [37]. Fat accumulation in obesity leads to the hypertrophy of adipocytes, putting cells in a state of stress and activating pro-inflammatory pathways. In addition, excessive enlargement of adipocytes leads to increased lipolysis, and the formation of metabolic endotoxemia (increased lipopolysaccharide and endotoxin in circulation) also contributes to inflammation [38,39]. The inflammatory factor TNF-α induces the decomposition of adipocytes to release free fatty acids, which in turn bind to Toll-like receptors on the surfaces of macrophages and adipocytes, further activating inflammatory pathway signals and promoting inflammatory factors release [40]. Furthermore, the transduction of insulin signals may be interfered with by inflammatory factors, resulting in insulin resistance, which then causes various metabolic disorders [41]. Polysaccharides have attracted the increasing attention of scientists around the world for their safety, potent anti-inflammatory, and immunomodulatory properties [42]. In our study, 50, 100, and 250 μg/mL of Polygonatum sibiricum polysaccharides alleviated IL‑1β, IL‑6, and TNF‑α levels and promoted proliferation, glucose uptake, and glucose transporter 4 expression in IR‑3T3‑L1 adipocytes [43]. Wu et al. reported that Ganoderma lucidum beta 1,3/1,6 glucan (MBG) suppressed high-cholesterol diet-induced inflammation in male 57BL/6J mice by stimulating serum IgA and IgG production, boosting poly-Ig receptor expression, and increasing IL-2 production by NK cells [44]. The results of our study demonstrate that AAP markedly down-regulated the gene expressions of IL-6, IL-1β, and Tnf-α, while up-regulating the gene expression of anti-inflammatory factor IL-10 in liver tissue (Figure 7), which could be one of the mechanisms underlying its liver-protecting and weight-reducing effects.

In recent years, studies have shown that the multiple biological activities of polysaccharides may be closely related to the regulation of intestinal flora. On the one hand, intestinal flora can degrade the polysaccharides to generate monosaccharides, thereby promoting the absorption and utilisation of the polysaccharides by the body, simultaneously generating metabolites with certain biological activity, such as short-chain fatty acids (SCFAs). On the other hand, the polysaccharides can directly increase the abundance of beneficial bacteria and reduce the abundance of harmful bacteria, improving the physiological levels of the body [45]. Zhao et al. compared the effects of six edible fungi (Auricular, Flammulina Velutipes, Lentinus edodes, Agaricus bisporus, Pleurotus ostertaus, and Pleurotus eryngii) on the composition and diversity of intestinal flora using an in vitro fermentation system. The results showed that edible fungi increased the content of SCFAs, thus reducing the pH of the fermentation broth, and had a significant effect on the diversity of intestinal flora. In particular, AAP could significantly increase the abundance of Bifidobacterium and Bacteroides and reduce the abundance of Fusobacterium [46]. Zhang et al. found that Auricularia auricula-judae powder and Auricularia auricula-judae crude polysaccharides both markedly reduced the levels of TC and LDL-C in model SD rats by regulating intestinal flora and its metabolites. Among them, Auricularia auricula-judae powder increased the abundance of Prevost's bacteria and high-abundance SCFAs-producing bacteroides related to a diet rich in dietary fibre, while Auricularia auricula-judae polysaccharides mainly promoted the abundance of low- abundance SCFAs -producing bacteria, such as Flavobacterium and Clostridium IV [11]. Although it has been demonstrated that many natural polysaccharides can improve the intestinal flora diversity of mice induced by a high-calorie diet, it is difficult to prove the causal relationships based on correlation analysis. The effective dose, molecular mechanisms, and metabolic process of polysaccharides in regulating obesity need to be systematically investigated.

The structure of an active substance determines its function. The types and contents of the active components in crops from different origins and varieties are different. Additionally, different extraction methods often affect the yield, structural properties, and biological activity of natural polysaccharides. At present, the research on the lipid-lowering of AAP mainly focuses on the well-known Northeast Auricularia auricula-judae resources. Our research team previously compared the physicochemical properties and antioxidant activities of AAP in the Northwest Qinba Mountains with AAP in Heilongjiang and analysed the differences between AAP obtained by different extraction methods, such as hot water extraction, enzymatic extraction, and ultrasonic extraction. The results showed that the AAP from the Qinba Mountains was mainly composed of mannose, glucuronic acid, and xylose, and the relative molar percentage was 60.87:20.83:9.86, which was similar to the northeast. In addition, the content of mannose and glucuronic acid in AAP prepared by hot water extraction was the highest. Mannose is not only an important monosaccharide for protein glycosylation in mammals but also an inefficient source of cellular energy. Sharma et al. found that a certain amount of mannose supplementation inhibited the weight gain of mice fed with a high-fat diet, as well as reduced the proportion of fat, prevented fatty liver, enhanced endurance and maximum oxygen consumption, and improved glucose tolerance [47]. Mannose increased the ratio of firmicutes to bacteroidetes in the gut microbiota of mice. Moreover, functional transcriptomic analysis of the mouse caecal microbiota revealed that the coherent changes in microbial energy metabolism were caused by mannose, and it is speculated that mannose played a role by reducing the energy produced by the metabolism of complex carbohydrates by intestinal flora and reducing energy intake [47]. The results of this study showed that dietary supplementation of 100 mg/kg·day of Auricularia auricula-judae water-extracted polysaccharide from the Qinba Mountains significantly reduced the food efficiency ratio, inhibited the accumulation of subcutaneous fat and epididymal fat, improved lipid metabolism, and slowed down the mice body weight gain induced by HFFD (Figure 1). Due to the complexity and diversity of polysaccharides structure, the relationship between molecular weight, monosaccharide composition, glycosidic bond connection mode, the advanced structure of polysaccharides and their functional activities is not clear enough, the main active groups of AAP which play a role in weight loss need further in-depth investigation.

Reviewer 2 Report

In this study, the authors investigated the anti-obesity effects of Auricularia auricula polysaccharide in HFFD-induced nutritional obesity in mice. It’s an interesting study, however, some questions need to be considered.

Comments      

  1. The method and duration of feeding with APP should be added.
  2. The purpose of using fructose, not glucose needs to be discussed in the text.
  3. Figure 6C needs higher magnification.
  4. The label in figure 5 is confusing. What do “HFFD & ND” and “HFFD+APP&HFFD” mean?
  5. The mechanism study is superficial. I don’t think the authors could draw some conclusions about the molecular mechanism. Please revise the text in the introduction and discussion.
  6. What’s the purity of the polysaccharide? Any total sugar quantification?
  7. Minor issues:
  • Double-check the name of “Auricularia auricula” and keep them consistent in the text.
  • “TNF-α” in the table of primer.
  • What does “Necking” mean in line 140?

Author Response

Reviewer: 2

In this study, the authors investigated the anti-obesity effects of Auricularia auriculapolysaccharide in HFFD-induced nutritional obesity in mice. It’s an interesting study, however, some questions need to be considered.

Responses:

Thank you for your encouraging comments. We have studied comments carefully and revised our manuscript seriously. The manuscript has been polished by MDPI language editing services (https://www.mdpi.com/authors/english), and the revised portions are highlighted in blue in the revised manuscript. We hope that the corrections will meet with approval.

Comments:

  1. The method and duration of feeding with APP should be added.

Responses:

Thanks for the reviewer’s kind remind. The specific animal treatment methods in this research were as follows (described in revised manuscript Page 6, Line 131-137): the mice in the ND group were fed with AIN93M standard feed and distilled water. The mice in the HFFD induction group were fed with 45% high-fat feed and 10% high fructose drinking water. Mice in ND+AAP negative control group were given standard feed containing 200 mg/kg/day AAP and distilled water. HFFD+AAP intervention group mice were given 50, 100 or 200 mg/kg/day containing 45% high-fat feed of AAP and 10% high-fructose drinking water. The experimental period was 12 weeks. We have added animal experimental scheme (Figure 1 A) to make the processing clearer and more visual.

Figure 1.A Timeline of C57BL/6J mice with AAP and/or HFFD treatment

Comments:

  1. The purpose of using fructose, not glucose needs to be discussed in the text.

Responses:

Thanks for the reviewer’s kind remind. The comment is very helpful to improve the quality of discussion part. We have added some discussions about the purpose of using fructose in revised manuscript, please see in Page 23, Line 440-462.

Fructose is the sweetest monosaccharide and a natural sweetener, which is widely found in fruits, juices, some vegetables, and honey. Fructose is the favorite source of sweetness for most people. Structurally, fructose is a levorotatory hexulose and an isomer of glucose. Animal studies have shown that a high fructose diet can cause dyslipidemia, increased blood glucose, abdominal obesity, and increased blood pressure in rats, showing a more typical metabolic syndrome. Epidemiological studies have found that high fructose intake is associated with overweight, dyslipidemia and metabolic syndrome in the population 1. The harm of high fructose diet has been gradually recognized. A systematic review of the relationship between fructose and body weight demonstrated that moderate fructose intake in place of other carbohydrates will not lead to weight gain, whereas high fructose (> 100 g/ day) intake will result in increased body weight 2.

The fructose metabolism of the body is different from that of glucose in many aspects 3, which is specifically reflected in the following aspects:

Fructose absorption 4: Compared with glucose, fructose enters the blood at a slower rate and at a lower level with a longer half-life. Fructose is transported to small intestinal epithelial cells via glucose transporter 5 (GLUT5) and then spreads into the bloodstream through glucose transporter 2 (GLUT2). Compared with glucose, the absorption process requires neither ATP nor sodium. Due to the limitation of GLUT5, the total amount of fructose that can be absorbed by each person per day is limited. However, free fatty acid (FFA) and some amino acids can promote the absorption of fructose, and long-term intake of fructose will up-regulate the intestinal GLUT5 expressions 5. In addition, fructose can promote the binding of FFA and apolipoprotein ApoB48 in the small intestine, thereby improving the level of chylomicrons released by it in the blood, that is, improving the absorption of FFA; (2) the upregulation of GLUT5 expression in the small intestine enhances the body's ability to absorb fructose, that is, the total calories ingested from food are increased.

Fructose can promote food intake: Women with normal weight had lower levels of insulin and leptin secretion and higher levels of the gastrointestinal hormone ghrelin secretion after consuming fructose drinks compared to isocaloric glucose beverages 6. Fructose also promotes food intake by activating the dopamine signal transduction pathway in the nucleus accumbens outer shell of the mesencephalic limbic system. Moreover, KHK, a key enzyme for fructose metabolism, is expressed in cerebellum, hippocampus, cortex, and olfactory bulb, etc. The intermediate product of fructose metabolism in brain can inhibit the generation of malonyl coenzyme A through AMPK/ malonyl coenzyme A signal transduction system, so that the inhibitory effect of malonyl coenzyme A on neuropeptide system regulating appetite in hypothalamus can be alleviated, and then food intake can be further promoted 4.

Fructose slows down resting energy metabolism: Studies by Cox et al. have shown that compared with the intake of equal-calorie glucose drinks, overweight or obese subjects have significantly lower energy expenditure in the resting state after long-term intake of fructose drinks that account for 25% of the total energy, indicating that fructose can slow down energy metabolism in the resting state 7.

Fructose can directly promote fat synthesis 8: In the liver, fructose is catalyzed by KHK to produce 1-phosphate fructose with adenosine triphosphate (ATP) as the phosphoric acid donor. Fructose 1-phosphate is decomposed into dihydroxyacetone phosphate and glyceraldehyde under the catalytic action of aldolase B. Among them, dihydroxyacetone phosphate can form a glycerol backbone of triglycerides and phospholipids, while glyceraldehyde, under the catalytic action of propionose kinase phosphate, uses ATP as the phosphoric acid donor to generate the intermediate product of glycolysis 3-phosphoglyceride, which enters the glycolysis pathway to generate pyruvate and finally into acetyl-CoA. As fructose metabolism does not go through the rate-limiting reaction catalyzed by phosphofructokinase in glycolytic pathway and less regulated by the cell energy state, fructose can generate a large amount of acetyl-CoA, beyond the metabolic capacity of the mitochondrial tricarboxylic acid cycle, then excess acetyl-CoA enters the DNL pathway to synthesize fat. DNL is a metabolic pathway that converts excess non-fatty energy into fat by synthesizing fatty acids from acetyl-CoA.

Fructose promotes fat accumulation by inducing uric acid production 9,10: Another metabolic feature of fructose metabolism different from glucose metabolism is that fructose can induce uric acid production. Under the catalysis of KHK, fructose in hepatocytes uses ATP as a phosphate donor to produce fructose 1- phosphate and adenosine diphosphate (ADP). This KHK-catalyzed reaction progressed rapidly without negative feedback regulation, resulting in marked decreases in intracellular phosphate and ATP levels. The decreased intracellular phosphate activates AMP deaminase 2 (AMPD2), which catalyzes the degradation of adenosine monophosphate (AMP) into inosine monophosphate (IMP), which undergoes a series of enzymatic reactions to produce the final product uric acid.

Fructose can also activate the purine de novo biosynthesis pathway, and promote the generation of uric acid from amino acid precursors such as glycine. In addition, fructose can also inhibit uric acid excretion in kidney and ileum and increase uric acid level. Compared with equal calorie glucose drinks, the serum uric acid level of overweight or obese patients increased significantly after long-term intake of high fructose drinks for 10 weeks 11. Moreover, in the side-chain reaction of fructose metabolism to uric acid, the downstream end product uric acid catalyzed by AMPD2 can also significantly reduce AMPK activity, thereby reducing fatty acid oxidation and promoting fructose-induced fat accumulation. Acyl coenzyme A hydratase 1 is the rate-limiting enzyme of fatty acid β-oxidation. Uric acid can reduce the expression of acyl coenzyme A hydratase 1, reduce fatty acid oxidation and promote fat accumulation. Besides reducing fatty acid oxidation, uric acid can activate the pathway of fat synthesis by inducing mitochondrial oxidative stress and promote fat accumulation 12.

References

  1. Tappy, L.; Le, K. A., Metabolic effects of fructose and the worldwide increase in obesity. Physiol Rev 2010, 90 (1), 23-46.
  2. Sievenpiper, J. L.; de Souza, R. J.; Mirrahimi, A.; Yu, M. E.; Carleton, A. J.; Beyene, J.; Chiavaroli, L.; Di Buono, M.; Jenkins, A. L.; Leiter, L. A.; Wolever, T. M.; Kendall, C. W.; Jenkins, D. J., Effect of fructose on body weight in controlled feeding trials: a systematic review and meta-analysis. Annals of internal medicine 2012, 156 (4), 291- 304.
  3. Kanarek, R. B.; Orthen-Gambill, N., Differential effects of sucrose, fructose and glucose on carbohydrate-induced obesity in rats. The Journal of nutrition 1982, 112 (8), 1546-54.
  4. Helsley, R. N.; Moreau, F.; Gupta, M. K.; Radulescu, A.; DeBosch, B.; Softic, S., Tissue-Specific Fructose Metabolism in Obesity and Diabetes. Current diabetes reports 2020, 20 (11), 64.
  5. Johnson, R. J.; Perez-Pozo, S. E.; Sautin, Y. Y.; Manitius, J.; Sanchez-Lozada, L. G.; Feig, D. I.; Shafiu, M.; Segal, M.; Glassock, R. J.; Shimada, M.; Roncal, C.; Nakagawa, T., Hypothesis: Could Excessive Fructose Intake and Uric Acid Cause Type 2 Diabetes? Endocr Rev 2009, 30 (1), 96-116.
  6. Teff, K. L.; Elliott, S. S.; Tschop, M.; Kieffer, T. J.; Rader, D.; Heiman, M.; Townsend, R. R.; Keim, N. L.; D'Alessio, D.; Havel, P. J., Dietary fructose reduces circulating insulin and leptin, attenuates postprandial suppression of ghrelin, and increases triglycerides in women. The Journal of clinical endocrinology and metabolism 2004, 89 (6), 2963-72.
  7. Cox, C. L.; Stanhope, K. L.; Schwarz, J. M.; Graham, J. L.; Hatcher, B.; Griffen, S. C.; Bremer, A. A.; Berglund, L.; McGahan, J. P.; Havel, P. J.; Keim, N. L., Consumption of fructose-sweetened beverages for 10 weeks reduces net fat oxidation and energy expenditure in overweight/obese men and women. European journal of clinical nutrition 2012, 66 (2), 201-8.
  8. Hernandez-Diazcouder, A.; Romero-Nava, R.; Carbo, R.; Sanchez-Lozada, L. G.; Sanchez-Munoz, F., High Fructose Intake and Adipogenesis. International journal of molecular sciences 2019, 20 (11).
  9. Mortera, R. R.; Bains, Y.; Gugliucci, A., Fructose at the crossroads of the metabolic syndrome and obesity epidemics. Front Biosci-Landmrk 2019, 24, 186-211.
  10. King, C.; Lanaspa, M. A.; Jensen, T.; Tolan, D. R.; Sanchez-Lozada, L. G.; Johnson, R. J., Uric Acid as a Cause of the Metabolic Syndrome. Contrib Nephrol 2018, 192, 88-102.
  11. Cox, C. L.; Stanhope, K. L.; Schwarz, J. M.; Graham, J. L.; Hatcher, B.; Griffen, S. C.; Bremer, A. A.; Berglund, L.; McGahan, J. P.; Keim, N. L.; Havel, P. J., Consumption of fructose- but not glucose-sweetened beverages for 10 weeks increases circulating concentrations of uric acid, retinol binding protein-4, and gamma-glutamyl transferase activity in overweight/obese humans. Nutr Metab 2012,

12. Sanchez-Lozada, L. G.; Andres-Hernando, A.; Garcia-Arroyo, F. E.; Cicerchi, C.; Li, N.; Kuwabara, M.; Roncal-Jimenez, C. A.; Johnson, R. J.; Lanaspa, M. A., Uric acid activates aldose reductase and the polyol pathway for endogenous fructose and fat production causing development of fatty liver in rats. The Journal of biological chemistry 2019, 294 (11), 4272-4281.

Comments:

  1. Figure 6C needs higher magnification.

Responses:

Thanks for the reviewer’s kind remind. We have updated Figure 6C with higher magnification (×400).

Figure 6 (C) Representative H&E staining of liver (×400).

Comments:

  1. The label in figure 5 is confusing. What do “HFFD & ND” and “HFFD+APP&HFFD” mean?

Responses:

Thanks for the reviewer’s kind remind. In Figure 5, “HFFD & ND” means the differential metabolites of “HFFD group compared with ND group”, and “HFFD+AAP&HFFD” means the differential metabolites of “HFFD+AAP group compared with HFFD group”. We have corrected the legend descriptions in Figure 5 to make it more clearly.

Comments:

  1. The mechanism study is superficial. I don’t think the authors could draw some conclusions about the molecular mechanism. Please revise the text in the introduction and discussion.

Responses:

Thanks for the reviewer’s kind remind. Molecular mechanism refers to the relationships between the structural components of organisms, as well as the physical and chemical properties and the interrelation of various changes in this process. Researchers usually focus on the "DNA-RNA-protein" central principle to explore the molecular mechanisms of active substances. In this study, differential metabolites related to lipids were screened out from obese mice after intake of AAP, and the mechanisms of lipid- lowering and weight-loss of AAP were attempted to be elucidated from many aspects, such as genes related to lipid synthesis, genes related to lipid decomposition, genes related to cholesterol synthesis and efflux. In-depth potential molecular mechanisms will be further investigated from the post-transcriptional levels and cellular signaling pathways. We have revised the introduction and discussion parts, please see in Page 3 and Page 22-28 in the revised manuscript.

In the discussion parts, we discussed the mechanism of polysaccharide's functional activity from the aspects of genes related to lipid metabolism, inflammation, intestinal flora, etc. We also compared with other polysaccharides, specifically as follows:

【Numerous studies have shown that natural polysaccharides can improve the disorder of lipid metabolism by changing genes related to lipid metabolism. Polysaccharides regulate TG metabolism mainly through three pathways, the ATGL-(PPAR-α)-(PGC-1α) pathway, the (SREBP-1c)-ACC/FAS pathway, and the ACC-CPT1 pathway, and regulate cholesterol metabolism primarily through the (SREBP-2)-HMGCR pathway, the PCSK9-LDLR pathway, and bile acid synthesis pathway [30]. Wu et al. screened out 13 genes involved in lipid metabolism in liver and epididymal tissue by gene expression profiling array and found that black tea polysaccharides inhibited fat formation, accelerated fat digestion, and promoted lipolysis by regulating the expressions of differential genes affecting lipid metabolism, including bile acid secretion, transforming growth factor signalling, insulin signalling, glycolipid metabolism, fatty acid degradation, and the AMPK signalling pathway [31]. The polysaccharide from Cyclocarya paliurus leaves exerts therapeutic effects on hyperlipidaemic rats through the induction of ATGL and PPAR-α and down-regulation of FAS and HMGCR [32]. Schisandra polysaccharide has been shown to markedly suppress the hepatic lipogenesis related genes, SREBP-1c, ACC, and FAS expressions [33]. The intervention of 800 mg/kg/d green tea polysaccharides could up-regulate the expressions of CPT-1, down-regulate the expressions of PPARγ, SREBP-1c, FAS, and LXRα, thus significantly reducing the fat index and adipocyte area, and inhibiting mice obesity [34,35]. In this study, we found that AAP down-regulated the mRNA expressions of liver adipogenic genes Pparg, Srebp1c, Fasn, Acaca, and ME1, suppressed cholesterol synthesis related genes Hmgcr and Acat2 mRNA levels, up-regulated fatty acid β-oxidation related genes Cpt1α and Cpt2 expressions, and promoted the expressions of cholesterol efflux related genes LXR, Abcg5, Abcg8, thus improving the homeostasis of liver lipid metabolism in mice (Figure 8). Furthermore, it has been demonstrated that PGC- 1α plays an important role in a variety of metabolic processes, including mitochondrial biogenesis and mitochondrial β-oxidation, and PGC-1α has emerged as an important therapeutic target for fatty liver disease [36]. AAP up-regulated Pgc1α and Sirt1 expressions, and its intervention on HFFD-induced liver lipid metabolism disorder was closely related to the regulatory effects on mitochondrial function. However, as a non-digestible polysaccharide, how AAP acts on the target tissues and exerts its anti-obesityeffect needs further in-depth study.

The body is in a state of chronic inflammation when obesity occurs, and the levels of TNF-α, ILs, and other inflammatory factors in the circulation and tissues of obese patients are significantly increased [37]. Fat accumulation in obesity leads to the hypertrophy of adipocytes, putting cells in a state of stress and activating pro-inflammatory pathways. In addition, excessive enlargement of adipocytes leads to increased lipolysis, and the formation of metabolic endotoxemia (increased lipopolyaccharide and endotoxin in circulation) also contributes to inflammation [38,39]. The inflammatory factor TNF-α induces the decomposition of adipocytes to release free fatty acids, which in turn bind to Toll-like receptors on the surfaces of macrophages and adipocytes, further activating inflammatory pathway signals and promoting inflammatory factors release [40]. Furthermore, the transduction of insulin signals may be interfered with by inflammatory factors, resulting in insulin resistance, which then causes various metabolic disorders [41]. Polysaccharides have attracted the increasing attention of scientists around the world for their safety, potent anti-inflammatory, and immunomodulatory properties [42]. In our study, 50, 100, and 250 μg/mL of Polygonatum sibiricum polysaccharides alleviated IL‑1β, IL‑6, and TNF‑α levels and promoted proliferation, glucose uptake, and glucose transporter 4 expression in IR‑3T3‑L1 adipocytes [43]. Wu et al. reported that Ganoderma lucidum beta 1,3/1,6 glucan (MBG) suppressed high-cholesterol diet-induced inflammation in male C57BL/6J mice by stimulating serum IgA and IgG production, boosting poly-Ig receptor expression, and increasing IL-2 production by NK cells [44]. The results of our study demonstrate that AAP markedly down-regulated the gene expressions of IL-6, IL-1β, and Tnf-α, while up-regulating the gene expression of anti-inflammatory factor IL-10 in liver tissue (Figure 7), which could be one of the mechanisms underlying its liver-protecting and weight-reducing effects.

In recent years, studies have shown that the multiple biological activities of polysaccharides may be closely related to the regulation of intestinal flora. On the one hand, intestinal flora can degrade the polysaccharides to generate monosaccharides, thereby promoting the absorption and utilisation of the polysaccharides by the body, simultaneously generating metabolites with certain biological activity, such as short-chain fatty acids (SCFAs). On the other hand, the polysaccharides can directly increase the abundance of beneficial bacteria and reduce the abundance of harmful bacteria, improving the physiological levels of the body [45]. Zhao et al. compared the effects of six edible fungi (Auricular, Flammulina Velutipes, Lentinus edodes, Agaricus bisporus, Pleurotus ostertaus, and Pleurotus eryngii) on the composition and diversity of intestinal flora using an in vitro fermentation system. The results showed that edible fungi increased the content of SCFAs, thus reducing the pH of the fermentation broth, and had a significant effect on the diversity of intestinal flora. In particular, AAP could significantly increase the abundance of Bifidobacterium and Bacteroides and reduce the abundance of Fusobacterium [46]. Zhang et al. found that Auricularia auricula-judae powder and Auricularia auricula-judae crude polysaccharides both markedly reduced the levels of TC and LDL-C in model SD rats by regulating intestinal flora and its metabolites. Among them, Auricularia auricula-judae powder increased the abundance of Prevost's bacteria and high-abundance SCFAs-producing bacteroides related to a diet rich in dietary fibre, while Auricularia auricula-judae polysaccharides mainly promoted the abundance of low- abundance SCFAs -producing bacteria, such as Flavobacterium and Clostridium IV [11]. Although it has been demonstrated that many natural polysaccharides can improve the intestinal flora diversity of mice induced by a high-calorie diet, it is difficult to prove the causal relationships based on correlation analysis. The effective dose, molecular mechanisms, and metabolic process of polysaccharides in regulating obesity need to be systematically investigated.

The structure of an active substance determines its function. The types and contents of the active components in crops from different origins and varieties are different. Additionally, different extraction methods often affect the yield, structural properties, and biological activity of natural polysaccharides. At present, the research on the lipid-lowering of AAP mainly focuses on the well-known Northeast Auricularia auricula-judae resources. Our research team previously compared the physicochemical properties and antioxidant activities of AAP in the Northwest Qinba Mountains with AAP in Heilongjiang and analysed the differences between AAP obtained by different extraction methods, such as hot water extraction, enzymatic extraction, and ultrasonic extraction. The results showed that the AAP from the Qinba Mountains was mainly composed of mannose, glucuronic acid, and xylose, and the relative molar percentage was 60.87:20.83:9.86, which was similar to the northeast. In addition, the content of mannose and glucuronic acid in AAP prepared by hot water extraction was the highest. Mannose is not only an important monosaccharide for protein glycosylation in mammals but also an inefficient source of cellular energy. Sharma et al. found that a certain amount of mannose supplementation inhibited the weight gain of mice fed with a high-fat diet, as well as reduced the proportion of fat, prevented fatty liver, enhanced endurance and maximum oxygen consumption, and improved glucose tolerance [47]. Mannose increased the ratio of firmicutes to bacteroidetes in the gut microbiota of mice. Moreover, functional transcriptomic analysis of the mouse caecal microbiota revealed that the coherent changes in microbial energy metabolism were caused by mannose, and it is speculated that mannose played a role by reducing the energy produced by the metabolism of complex carbohydrates by intestinal flora and reducing energy intake [47]. The results of this study showed that dietary supplementation of 100 mg/kg·day of Auricularia auricula-judae water-extracted polysaccharide from the Qinba Mountains significantly reduced the food efficiency ratio, inhibited the accumulation of subcutaneous fat and epididymal fat, improved lipid metabolism, and slowed down the mice body weight gain induced by HFFD (Figure 1). Due to the complexity and diversity of polysaccharides structure, the relationship between molecular weight, monosaccharide composition, glycosidic bond connection mode, the advanced structure of polysaccharides and their functional activities is not clear enough, the main active groups of AAP which play a role in weight loss need further in-depth investigation】

Comments:

  1. What’s the purity of the polysaccharide? Any total sugar quantification?

Responses:

Thanks for the reviewer’s kind remind. In this study, Auricularia auricula polysaccharides were prepared by hot water extraction and alcohol precipitation method. According to the existing methods of our research group (Food Chemistry, 2020; Carbohydrate Polymers, 2019), phenol-sulfuric acid method was used to determine the sugar content, BCA method was used to determine the protein content, and sulfuric acid- carbazole method was used to determine the uronic acid content in the polysaccharide samples. The carbohydrate content, protein content and uronic acid content of AAP were 53.15±4.52 %, 1.45±0.12 % and 11.04±0.59 %, respectively. We have added the related physicochemical properties descriptions of AAP in the experimental materials parts, please see Page 5, line 114-116

Comments:

  1. Minor issues:
  • Double-check the name of “Auricularia auricula” and keep them consistent in the text.
  • “TNF-α” in the table of primer.
  • What does “Necking” mean in line 140?

Responses:

Thanks for the reviewer’s kind remind.

We have double-checked the name of “Auricularia auricula” and uniformed them to “Auricularia auricula-judae” in the revised manuscript.

We have corrected the primer “Tnfα” in revised manuscript, please see Page 9.

“Necking” means “killed by cervical dislocation” after anesthesia. We have corrected the animal treatment description, please see Page 7, line 159.

Reviewer 3 Report

The manuscript “Dietary supplementation of Auricularia auricula polysaccharides alleviate nutritional obesity in mice via regulating inflammatory response and lipid metabolism” provides interesting and practical data on the polysaccharides extract of this mushroom. I consider that this paper is one of the first approaches in this field and fits into scope of this scientific journal; the manuscript, however, requires a lot of corrections and an extensive editing of English language.

General comments

  1. The section "materials and methods" is described in several parts in a schematic and not very discursive way.
  2. In the "Introduction" section, mention could be made of other natural extracts tested for similar purposes (e.g.van der Zande, H., Lambooij, J. M., Chavanelle, V., Zawistowska-Deniziak, A., Otero, Y., Otto, F., Lantier, L., McGuinness, O. P., Le Joubioux, F., Giera, M., Maugard, T., Peltier, S. L., Sirvent, P., & Guigas, B. (2021). Effects of a novel polyphenol-rich plant extract on body composition, inflammation, insulin sensitivity, and glucose homeostasis in obese mice. International journal of obesity (2005)45(9), 2016–2027. https://doi.org/10.1038/s41366-021-00870-x)

Lines 13,16,24, 240, 267, 290, 317, 334, 348, 594, 596, 598, 601, 602, 605, 607, 629: Species’ names should be in italics.

Line 82,83: capitalize the first letter of the generus'name.

Line 86: lowercase the first letter of the species’ name

Line 87-90: Materials and Methods”: This part is described as schematic steps. Please make the speech more discursive.

Lines 97-99: Did the composition of Auricularia auricular-judae evaluate in this study? However, “materials and methods” is not the correct place where describe the composition.

Line 112: What does “ND” stand for? Before using abbreviations, please use the full name.

Line 117: this sentence can be improved to explain better the treated groups.

Lines 118-119: “Make every effort…” Please make the speech more discursive.

Lines 122-123: Please correct this sentence as it cannot start with "And calculate" without subject.

Lines: 133-135: Please make the speech more discursive and clear.

Line 135: You could improve the sentence by using “ We used the following formula to calculate…”

Line 139: It’s not possible to collect blood sample from “the eyeballs”. You can use “blood samples were collected from retro-orbital sinus/plexus”

Line 143: Please make this sentence discursive. For example: “ We calculated the liver idex…” or “We used the following formula…”

Line 176: lowercase the first letter of “These”.

Lines 179-183: Write the sentences in correct English and more discursive.

Line 198: “Eache experiment was repeated at least six TIME, and THE OBTAINED DATA were expressed …”

Line 223: Please, write only the name of the mushroom in italics.

Line 229: “every week AND energy intake

Line 257: Better not to start the sentence with "and"

Line 273:Is  it a result of this study? Otherwise insert the citation.

Lines 277-284: This part should be put into "discussions" section. However, please insert citations.

Lines 305-309: This part is not clear.  Can you clarify it?

Line 315: “ AND biosynthesis of unsaturated fatty acids and steroid hormone.” (please delete the last “biosynthesis”.)

Line 321:Please insert citation.

Line 324: please delete “all”

Line 427: please insert citation.

Line 428: Change “ratio of Bacteroides to Firmicutes” in “ratio of Firmicutes to Bacteroidetes”

Line 438: “Eating habits” instead of “foodborne obesity”

Lines 445-447: Please, write this sentence more clearly and without repetition.

Discussion: Lines 466- 470 Insert citations.

Line 486: delete the repetition (fatty acid transporters)

Line 553: Please insert citation.

References: Lines 625, 637, 645, 647, 655, 666, 670, 672, 677, 679: Often the names of the authors are wrong. Check all references.

Author Response

Reviewer 3:

The manuscript “Dietary supplementation of Auricularia auricula polysaccharides alleviate nutritional obesity in mice via regulating inflammatory response and lipid metabolism” provides interesting and practical data on the polysaccharides extract of this mushroom. I consider that this paper is one of the first approaches in this field and fits into scope of this scientific journal; the manuscript, however, requires a lot of corrections and an extensive editing of English language.

Responses:

Thank you for your encouraging comments. We have studied comments carefully and revised our manuscript seriously. The manuscript has been polished by MDPI language editing services (https://www.mdpi.com/authors/english), and the revised portions are highlighted in blue in the revised manuscript. We hope that the corrections will meet with approval.

Comments:

  1. The section "materials and methods" is described in several parts in a schematic and not very discursive way.

Responses:

Thanks for the reviewer’s kind remind. We have re-constructed the "materials and methods" section (mainly consist of 2.1 Samples Preparation, 2.2 Experimental Design, 2.3 Assessments) in the revised manuscript, and added animal experimental scheme (Figure 1 A) to make the processing clearer and more visual.

Figure 1.A Timeline of C57BL/6J mice with AAP and/or HFFD treatment

Comments:

  1. In the "Introduction" section, mention could be made of other natural extracts tested for similar purposes (e.g.van der Zande, H., Lambooij, J. M., Chavanelle, V., Zawistowska-Deniziak, A., Otero, Y., Otto, F., Lantier, L., McGuinness, O. P., Le Joubioux, F., Giera, M., Maugard, T., Peltier, S. L., Sirvent, P., & Guigas, B. (2021). Effects of a novel polyphenol-rich plant extract on body composition, inflammation, insulin sensitivity, and glucose homeostasis in obese mice. International journal of obesity (2005), 45(9), 2016–2027. https://doi.org/10.1038/s41366-021-00870-x)

Responses:

Thanks for the reviewer’s valuable advice. We have added some descriptions about natural plant extracts which have potent anti-obesity activities, and added the reference [4] (van der Zande, H. J. P.; Lambooij, J. M.; Chavanelle, V.; Zawistowska-Deniziak, A.; Otero, Y.; Otto, F.; Lantier, L.; McGuinness, O. P.; Le Joubioux, F.; Giera, M.; Maugard, T.; Peltier, S. L.; Sirvent, P.; Guigas, B., Effects of a novel polyphenol-rich plant extract on body composition, inflammation, insulin sensitivity, and glucose homeostasis in obese mice. International journal of obesity 2021, 45 (9), 2016-2027.) in the "Introduction" section, please see in Page 3, Line 47-49.

Comments:

  1. Lines 13,16,24, 240, 267, 290, 317, 334, 348, 594, 596, 598, 601, 602, 605, 607, 629: Species’ names should be in italics.

Responses:

Thanks for the reviewer’s kind remind. We have italicized the Species’ names (“Auricularia auricula-judae”) in the revised manuscript.

Comments:

  1. Line 82,83: capitalize the first letter of the generus' name.

Responses:

Thanks for the reviewer’s kind remind. We have capitalized the first letter of the generus' name (“Auricularia auricula-judae”) in the revised manuscript, please see Page 5, line 90-91.

Comments:

  1. Line 86: lowercase the first letter of the species’ name

Responses:

Thanks for the reviewer’s kind remind. We have lowercased the first letter of the species’ name (“Auricularia auricula-judae”) in the revised manuscript.

Comments:

  1. Line 87-90: Materials and Methods”: This part is described as schematic steps. Please make the speech more discursive.

Responses:

Thanks for the reviewer’s kind remind. We have re-written the extraction procedure of AAP in detail, please see in Page 4-5, Line 92-116 in the revised manuscript. Specifically: 【Samples Preparation: The fruiting bodies of Auricularia auricula-judae (purchased from Shaanxi Tianmei Green Industry Co., Ltd. (Hanzhong, China), and grown in the Qinba Mountain area of northwest China) were crushed with a multi-functional crusher (200y, Yongkang Boou Hardware Products Co., Ltd., Yongkang, China) and sieved by 40-mesh sieve. Auricularia auricula-judae powder (30 g) was extracted with boiling water (material–liquid ratio was 1:50) at 100 °C for 2 h. After centrifugation at 2808× g for 15 min, the supernatant was transferred into a fresh tube, and the precipitate was re-extracted. The supernatants were combined and concentrated using a rotary evaporator (R-1005, Zhengzhou Changcheng Science Technology and Trade Co., Ltd., Zhengzhou, China). Then, anhydrous ethanol was added into the concentrated solution in a ratio of 1:4 (volume fraction) for ethanol precipitation for 12 h. After centrifugation at 2808× g for 15 min, the supernatant was discarded, and the residue was retained and dissolved in hot water. After the residue was fully dissolved, the protein was removed by the Sevage method. Specifically, Sevage reagent (dichloromethane: n-butanol = 4:1, v/v) was added into the solution and stirred for 20 min to cause the protein to fully denature and precipitate out. After centrifugation at 2808× g for 15 min, the supernatant was transferred to an aqueous solution, and the precipitation layer and organic layer were discarded. The deproteinization procedure was repeated 4~5 times. The cellophane with a molecular cut-off of 3000 Da was used for dialysis in a flowing tap water for 3 days. Then, the solution was concentrated by a rotary evaporator, freeze-dried, and finally weighed and bagged to obtain a crude polysaccharide sample of Auricularia auricula-judae. The carbohydrate content, protein content, and uronic acid content of AAP were 53.15 ± 4.52%, 1.45 ± 0.12%, and 11.04 ± 0.59%, respectively. The AAP was mainly composed of mannose, glucuronic acid, xylose, and the relative mole percentage was 60.87: 20.83: 9.86.】

Comments:

  1. Lines 97-99: Did the composition of Auricularia auricular-judae evaluate in this study? However, “materials and methods” is not the correct place where describe the composition.

Responses:

Thanks for the reviewer’s kind remind. The physicochemical properties of

Auricularia auricula-judae polysaccharides were characterized previously (LIU Qian, ZHANG Xinyu, WANG Yan, WANG Anyu, ZHAO Yi, LONG Zixian, LIU Sining, DIE Yun, HUANG Linjuan. Physicochemical properties and antioxidant activities in vitro of Auricularia auricula polysaccharides in Qinba mountain area,2021,47(23):91- 97.DOI:10.13995/j.cnki.11-1802/ts.027141.). This study was performed to explore the intervention effects of polysaccharides extracted from Auricularia auricula-judae resources in the Qinba Mountain area on nutritional obesity in C57BL/6J mice induced by high fat and high fructose diets (HFFD), and to investigate their underlying molecular mechanisms. Therefore, we added a supplementary description of polysaccharides in the materials and methods sections to further analyze the potential mechanism.

Comments:

  1. Line 112: What does “ND” stand for? Before using abbreviations, please use the full name.

Responses:

Thanks for the reviewer’s kind remind. “ND” stands for “Normal diet”, and we have added the full name in the revised manuscript, please see Page 6, line 132.

Comments:

  1. Line 117: this sentence can be improved to explain better the treated groups.

Responses:

Thanks for the reviewer’s kind remind. We are polishing our manuscript through MDPI language editing services. And this sentence has been improved to “HFFD+AAP treated group mice were given 50, 100 or 200 mg/kg/day AAP in 45% high-fat feed respectively, and 10% high-fructose drinking water” in revised manuscript, please see Page 6, line 136.

Comments:

  1. Lines 118-119: “Make every effort...” Please make the speech more discursive.

Responses:

Thanks for the reviewer’s kind remind. The animal experiment protocol was approved by the Animal Ethics Committee of the Laboratory Animal Center of Northwest University (Approval Code: NWU-AWC-20200401M), and was carried out in accordance with the "Administrative Regulations on Laboratory Animals" of the National Science and Technology Commission of the People's Republic of China. We have added the Approval Code in the revised manuscript, and we ensure that our research complies with the commonly-accepted '3Rs'. All surgery was performed under anaesthesia, and all efforts were made to minimise suffering. And we have polished our manuscript through MDPI language editing services. (https://www.mdpi.com/authors/english).

Comments:

  1. Lines 122-123: Please correct this sentence as it cannot start with "And calculate"

without subject.

Responses:

Thanks for the reviewer’s kind remind. We have corrected this sentence (“The average food intake, energy intake and food efficiency ratio of the mice were calculated”) in the revised manuscript, please see Page 6, line 144.

Comments:

  1. Lines: 133-135: Please make the speech more discursive and clear.

Responses:

Thanks for the reviewer’s kind remind. We have corrected this sentence in the revised manuscript (“For the glucose tolerance test, glucose (2 g/kg body weight) was injected intraperitoneally, and plasma glucose levels were measured before and 15, 30, 60, and 120 min after the injection.”), please see Page 7, line 153-155.

Comments:

  1. Line 135: You could improve the sentence by using “ We used the following formula to calculate...”

Responses:

Thanks for the reviewer’s kind remind. We have improved this sentence in the revised manuscript (“We used the following formula to calculate the homeostasis model assessment of insulin resistance (HOMA-IR) index: fasting insulin concentration (mU/L) ×fasting glucose concentration (mg/dL) ×0.05551]/22.5.”), please see Page 7, line 153-157.

Comments:

  1. Line 139: It’s not possible to collect blood sample from “the eyeballs”. You can use “blood samples were collected from retro-orbital sinus/plexus”

Responses:

Thanks for the reviewer’s kind remind. We have corrected this sentence in the revised manuscript (“blood samples were collected from the retro-orbital sinus/plexus”), please see Page 7, line 160.

Comments:

  1. Line 143: Please make this sentence discursive. For example: “ We calculated the liver idex...” or “We used the following formula...”

Responses:

Thanks for the reviewer’s kind remind. We have improved this sentence in the revised manuscript (“We used the following formula to calculate Liver index (%): Liver index (%)=liver mass/body mass ×100%.”), please see Page 7, line 165.

Comments:

  1. Line 176: lowercase the first letter of “These”.

Responses:

Thanks for the reviewer’s kind remind. We have deleted “These” in the revised manuscript.

Comments:

  1. Lines 179-183: Write the sentences in correct English and more discursive.

Responses:

Thanks for the reviewer’s kind remind. We have corrected these sentences in the revised manuscript (“The tissue cell plasma was dyed with eosin staining solution for 3 min, then dehydrated with ethanol, and made transparent with xylene. Finally, the film was sealed with neutral resin and observed through an optical microscope.”), please see Page 8, line 204-206.

Comments:

  1. Line 198: “Eache experiment was repeated at least six TIME, and THE OBTAINED DATA were expressed ...”

Responses:

Thanks for the reviewer’s kind remind. We have improved this sentence in the revised manuscript (“Each experiment was repeated at least six times, and the obtained data were expressed as mean ±standard error ( ±SEM)”), please see Page 10, line 220.

Comments:

  1. Line 223: Please, write only the name of the mushroom in italics.

Responses:

Thanks for the reviewer’s kind remind. We have removed the inappropriate italics, please see Page 11, line 243 in the revised manuscript.

Comments:

  1. Line 229: “every week AND energy intake

Responses:

Thanks for the reviewer’s kind remind. We have corrected the sentence in the revised manuscript (“To further explore whether the increase in the weight of mice was caused by the increase in food intake, the energy intake and FER were calculated during the experiment period.”), please see Page 11, line 252-254.

Comments:

  1. Line 257: Better not to start the sentence with "and"

Responses:

Thanks for the reviewer’s kind remind. We have deleted "and" in the beginning of the sentence.x

Comments:

  1. Line 273: Is it a result of this study? Otherwise insert the citation.

Responses:

Thanks for the reviewer’s kind remind. We have removed inappropriate descriptions in the results presentation in the revised manuscript.

Comments:

  1. Lines 277-284: This part should be put into "discussions" section. However, please insert citations.

Responses:

Thanks for the reviewer’s kind remind. We have removed inappropriate discussions in the results presentation in the revised manuscript, please see Page 14.

Comments:

  1. Lines 305-309: This part is not clear. Can you clarify it?

Responses:

Thanks for the reviewer’s kind remind. We have improved the result descriptions to make it more clearly, please see Page 16, line 334-346 in the revised manuscript. We make an explanation as follows: We used mass spectrometry to detect lipids and lipid-related metabolites in mice serum.

In the positive ion mode (A), dietary supplement of AAP significantly increased the serum levels of 13 metabolites such as L-Palmitoylvanine in the serum of obese mice, and down-regulated 14 metabolites such as Glycerophosphocholine. Then, we analyzed the metabolic pathways of these metabolites through MetaboAnalyst database. It was found that these metabolites participated in Glycerophospholipid metabolism, Linolenic acid metabolism, alpha-Linolenic acid metabolism, Ether lipid metabolism, Arachidonic acid metabolism, Fatty acid degradation, and Steroid biosynthesis.

In addition, in the negative ion mode (B), 10 metabolites with significantly different changes, such as Dodecanoic acid, in the serum of mice were found. Metabolic pathways of these metabolites were analyzed through the MetaboAnalyst database, and they were found to be related to Fatty acid biosynthesis, Butanoate metabolism, Alanine, aspartate and glutamate metabolism, Biosynthesis of unsaturated fatty acids, and Steroid hormone biosynthesis.

Comments:

  1. Line 315: “ AND biosynthesis of unsaturated fatty acids and steroid hormone.” (please delete the last “biosynthesis”.)

Responses:

Thanks for the reviewer’s kind remind. We make an explanation as follows:

“Biosynthesis of unsaturated fatty acids” and “Steroid hormone biosynthesis” were two different metabolic pathways in the MetaboAnalyst database.

Comments:

  1. Line 321: Please insert citation.

Responses:

Thanks for the reviewer’s kind remind. We have added citation (Fabbrini, E.; Sullivan, S.; Klein, S., Obesity and nonalcoholic fatty liver disease: biochemical, metabolic, and clinical implications. Hepatology 2010, 51 (2), 679-89.) in the revised manuscript, please see reference [13].

Comments:

  1. Line 324: please delete “all”

Responses:

Thanks for the reviewer’s kind remind. We have deleted "all" in the revised manuscript.

Comments:

  1. Line 427: please insert citation.

Responses:

Thanks for the reviewer’s kind remind. We have added citation (Sharma, V.; Smolin, J.; Nayak, J.; Ayala, J. E.; Scott, D. A.; Peterson, S. N.; Freeze, H. H., Mannose Alters Gut Microbiome, Prevents Diet-Induced Obesity, and Improves Host Metabolism. Cell Reports 2018, 24 (12).) in the revised manuscript, please see reference [47].

Comments:

  1. Line 428: Change “ratio of Bacteroides to Firmicutes” in “ratio of Firmicutes to Bacteroidetes”

Responses:

Thanks for the reviewer’s kind remind. We have changed “ratio of Bacteroides to Firmicutes” in “ratio of Firmicutes to Bacteroidetes” in revised manuscript, please see Page 27, line 568.

Comments:

  1. Line 438: “Eating habits” instead of “foodborne obesity”

Responses:

Thanks for the reviewer’s kind remind. We have changed “Eating habits” instead of “foodborne obesity” in the revised manuscript, please see Page 23, line 440.

Comments:

  1. Lines 445-447: Please, write this sentence more clearly and without repetition.

Responses:

Thanks for the reviewer’s kind remind. As the discussion part has been greatly revised, this sentence has been deleted.

Comments:

  1. Discussion: Lines 466- 470 Insert citations.

Responses:

Thanks for the reviewer’s kind remind. We have added citation (Nakarai, H.; Yamashita, A.; Nagayasu, S.; Iwashita, M.; Kumamoto, S.; Ohyama, H.; Hata, M.; Soga, Y.; Kushiyama, A.; Asano, T.; Abiko, Y.; Nishimura, F., Adipocyte-macrophage interaction may mediate LPS-induced low-grade inflammation: potential link with metabolic complications. Innate immunity 2012, 18 (1), 164-70. Grisouard, J.; Bouillet, E.; Timper, K.; Radimerski, T.; Dembinski, K.; Frey, D. M.; Peterli, R.; Zulewski, H.; Keller, U.; Muller, B.; Christ-Crain, M., Both inflammatory and classical lipolytic pathways are involved in lipopolysaccharide-induced lipolysis in human adipocytes. Innate immunity 2012, 18 (1), 25-34.) in the revised manuscript, please see reference [38,39].

Comments:

  1. Line 486: delete the repetition (fatty acid transporters)

Responses:

Thanks for the reviewer’s kind remind. As the discussion part has been greatly revised, this sentence has been deleted.

Comments:

  1. Line 553: Please insert citation.

Responses:

Thanks for the reviewer’s kind remind. As the discussion part has been greatly revised, this sentence has been deleted.

Comments:

  1. References: Lines 625, 637, 645, 647, 655, 666, 670, 672, 677, 679: Often the names of the authors are wrong. Check all references.

Responses:

Thanks for the reviewer’s kind remind. We have checked all the references and corrected in the revised manuscript

Round 2

Reviewer 1 Report

I have carefully read the manuscript and the authors have made my suggestions. The materials and methods have been adequately explained and the discussion clearly improved (although not totally well).

For all these reasons, I consider that the article is relevant enough to be published in the journal.

Reviewer 2 Report

Thanks for the professional responses.